# How Do Transformers Learn to Associate Tokens: Gradient Leading Terms Bring Mechanistic Interpretability

**Shawn Im**[1]**, Changdae Oh**[1]**, Zhen Fang**[2]**, Sharon Li**[1]
[1]University of Wisconsin–Madison    [2]University of Technology Sydney
{shawnim,changdae,sharonli}@cs.wisc.edu, zhen.fang@uts.edu.au

## Abstract

Semantic associations such as the link between "bird" and "flew" are foundational for language modeling as they enable models to go beyond memorization and instead generalize and generate coherent text. Understanding how these associations are learned and represented in language models is essential for connecting deep learning with linguistic theory and developing a mechanistic foundation for large language models. In this work, we analyze how these associations emerge from natural language data in attention-based language models through the lens of training dynamics. By leveraging a leading-term approximation of the gradients, we develop closed-form expressions for the weights at early stages of training that explain how semantic associations first take shape. Through our analysis, we reveal that each set of weights of the transformer has closed-form expressions as simple compositions of three basis functions–bigram, token-interchangeability, and context mappings–reflecting the statistics of the text corpus and uncovering how each component of the transformer captures semantic associations based on these compositions. Experiments on real-world LLMs demonstrate that our theoretical weight characterizations closely match the learned weights, and qualitative analyses further show how our theorem shines light on interpreting the learned associations in transformers.

## 1 Introduction

Large language models (LLMs) based on self-attention have shown strong capabilities in capturing both factual knowledge and qualitative aspects of the human world (Grattafiori et al., 2024; Yang et al., 2025; Team et al., 2024; Achiam et al., 2023). This progress has sparked growing interest in understanding why these models work so well and, in particular, what kinds of internal structures emerge during training (Engels et al., 2024; Li et al., 2023a; Meng et al., 2022; Cunningham et al., 2023). Among these structures, semantic associations are especially foundational to language modeling (Harris, 1954; Firth, 1957; Miller & Charles, 1991), as they enable models to connect words and concepts in ways that support generalization and coherent text generation. While recent studies have identified specific mechanisms such as induction heads (Olsson et al., 2022), linear semantic relations (Nanda et al., 2023), and topic clustering (Li et al., 2023b), we still lack a principled account of *how semantic associations arise during the training of attention-based transformers*.

By semantic associations, we mean the statistical and functional relationships between tokens that encode meaning—for example, the link between "bird" and "flew", the interchangeability of "car" and "truck" in adjectival contexts, or the coupling of "country" and "capital". These associations have long been recognized in linguistics under the lens of distributional semantics (Harris, 1954). In modern transformers, such associations are not explicitly programmed but instead emerge through gradient-based optimization over large corpora. Understanding *how* these structures crystallize during training is therefore essential not only for connecting deep learning with linguistic theory but also for developing a mechanistic foundation of representation learning in large language models.

In this work, we develop a theory for the emergence of semantic associations in attention-based language models trained on natural language data, through the lens of training dynamics. A formal

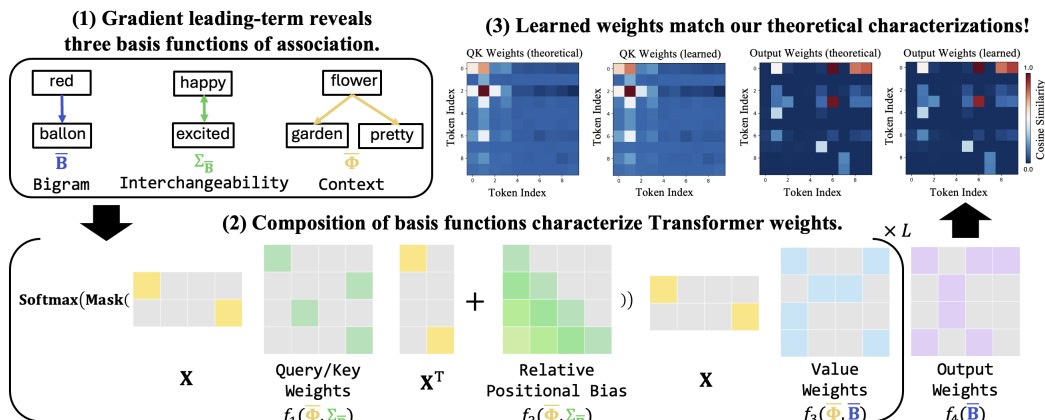

Figure 1: To understand the emergence of associative features, we analyze the training dynamics of Transformers by focusing on the gradient leading terms for weights, which allows us to identify interpretable basis functions that characterize each weight by their compositions. Empirical validation confirms that our weight characterizations match the actual ones learned in practical transformers.

analysis of training dynamics is attractive as it allows us to rigorously discuss how modern language models learn features and capabilities. Unfortunately, the training dynamics of transformers are highly complex, which has led prior work to adopt unrealistic assumptions that diverge from practice: (1) synthetic structured language (Li et al., 2023b; Yang et al., 2024), (2) simplified model architectures without, *e.g.*, positional encoding or residual connections (Tian et al., 2023; Huang et al., 2025), and (3) non-standard training, such as sequential component-wise training or partially frozen weights (Bietti et al., 2023; Li et al., 2023b). While these prior works provide valuable theoretical insights, their departures from realistic conditions raise concerns about the generalizability of their insights to LLMs used in practice. In contrast, we ground our study in a more realistic setting by focusing on naturalistic text distributions and attention-based transformers with positional encodings, optimized with a standard training procedure (Brown et al., 2020). This is essential to minimize the gap between our theory and practical use.

Our key technical innovation is to analyze training dynamics of the transformer at an early stage, through the leading term of an expansion of the gradients for each set of weights. In particular, transformers are known to acquire many core behaviors early in training–including semantic relations–and persist through convergence (Olsson et al., 2022; Elhage et al., 2021; Nanda et al., 2023). This makes the early phase not only empirically important but also analytically tractable. During this stage, gradient updates admit a closed-form approximation: the leading term dominates parameter updates before higher-order corrections accumulate. Leveraging this, we show that the learned weight matrices (including the output matrix, value matrices, query-key matrices) can be expressed as simple compositions of three basis functions: a *bigram mapping*, which captures next token dependencies; an *interchangeability mapping*, which reflects functional similarity across tokens (e.g., synonyms or shared grammatical roles); and a *context mapping*, which encodes longer-range prefix–suffix co-occurrence.

Through experiments on a natural language dataset, we verify that the learned weights in an attention-based transformer model closely match our theoretical closed-form expressions, and further demonstrate that this holds even beyond the early stage. We also show rich qualitative examples of how each weight component of the transformer captures the actual word-wise semantic associations characterized by our theorem. Furthermore, we verify that our theoretically characterized features are correlated with the behavior of real-world language model. Figure 1 depicts an overview of our analysis, and we summarize our **contributions** as follows:

1. We present the first explicit characterization of weights in attention-based transformers trained on real-world text corpora under the next-token prediction loss;

2. We interpret the features learned in weights as compositions of bi-gram, interchangeability, and context mappings, and then show how these basis functions capture semantic association across words;

3. We finally validate our theoretical interpretation on both self-attention models and practical LLM, demonstrating the generality and relevance of our theorems.

## 2 RELATED WORKS

**Understanding emergence of features in Transformers.**    Many works have considered the training dynamics of transformers under controlled settings to interpret their feature learning (Tian et al., 2023; Bietti et al., 2023; Nichani et al., 2024; Kim & Suzuki, 2024). A line of them investigates how low-level associative features, such as bigram structure (Bietti et al., 2023), cyclic structure (Huang et al., 2025), and co-occurrence (Tian et al., 2023; Yang et al., 2024), are learned from data. There are also multiple works that analyze how high-level capabilities, such as chain-of-thought (Kim & Suzuki, 2025), topic clustering (Li et al., 2023b; Jiang et al., 2024), reasoning or memorization (Yao et al., 2025), and in-context learning capability (Nichani et al., 2024; Bietti et al., 2023; Wang et al., 2024a; Kim & Suzuki, 2024; Edelman et al., 2024), are obtained during training. Although insightful, they often assume structured or abstract language data (Li et al., 2023b; Nichani et al., 2024; Yang et al., 2024), unrealistic model architecture (Tian et al., 2023; Cui et al., 2024; Troiani et al., 2025), and adjusted training strategies far from practice (Bietti et al., 2023; Kim & Suzuki, 2024; Huang et al., 2025), which depart from reality. In contrast, our theoretical analysis is grounded in natural language data, realistic architecture, and a standard training strategy. As a result, our theory substantially reduces the gap between formal analysis and practical use, which is further corroborated by our empirical validations.

**Understanding feature learning beyond Transformers.**    Recent work has also explored how models learn data-dependent features through dynamics for non-transformer models as well (Dandi et al., 2023; Ba et al., 2022; Mousavi-Hosseini et al., 2023). However, this line of work similarly considers abstractions of language, such as Gaussian data (Ba et al., 2022), single or multi-index models (Damian et al., 2024; Dandi et al., 2023), or spiked models (Wang et al., 2024b; Mousavi-Hosseini et al., 2023), and considers measures of data complexity with Hermite expansions (Bietti et al., 2022; Damian et al., 2024; Lee et al., 2024). On the contrary, we adopt a realistic theoretical setup to analyze features in transformers, which remains the dominant architecture in practice.

## 3 PRELIMINARY

### 3.1 PROBLEM STATEMENT

Semantic associations are foundational for language models: they enable models to go beyond memorizing sequences and instead generalize across contexts (Hinton, 1984), infer latent structure (Wu et al., 2018), and generate coherent text. Despite their importance, the mechanisms by which transformers acquire these associations during training remain poorly understood. Towards a *mechanistic and theory-grounded interpretation* of LLMs in a more realistic setup, we pose the question:

> *How do semantic associations emerge during the training of attention-based language models on natural language data?*

It is worth noting that we focus here on general natural language data, rather than synthetically structured or abstractive language, which has been considered in previous works (Yang et al., 2024; Nichani et al., 2024; Huang et al., 2025). This is essential to minimize the gap between our theory and practical use, since real-world text is highly diverse and is not restricted to a specific structure. In addition, prior studies (Olsson et al., 2022; Elhage et al., 2021; Nanda et al., 2023) have shown that critical semantic and reasoning abilities, such as induction heads and linear semantic relations, can already emerge in the early stage and be preserved through convergence. This makes the early stage of training a natural and necessary focus for theoretical analysis, which we now develop.

### 3.2 MODEL ARCHITECTURE

Prior works have analyzed the training dynamics of attention-based models under simplifying assumptions, such as restricting attention to low rank (Cui et al., 2024), removing causal masking (Tian et al., 2023; Yang et al., 2024), without positional encodings (Bietti et al., 2023) or residual

streams (Huang et al., 2025). In line with Nichani et al. (2024), we study an attention-based architecture that retains these components: positional encodings, causal masking, and residual streams. To further align with practice, we employ a relative positional encoding scheme, as in T5 (Raffel et al., 2020), rather than augmenting embeddings with absolute position vectors. We begin by introducing the necessary notation before formally defining the transformer computation.

Let $\mathcal{V} = \{\mathbf{e}_1, ..., \mathbf{e}_j, ..., \mathbf{e}_{|\mathcal{V}|}\}$ denote the set of vocabulary. For an input sequence of length $T$, we represent the input as a matrix $\mathbf{X} \in \mathbb{R}^{T \times |\mathcal{V}|}$, where each row of $\mathbf{X}$ is the one-hot encoding of the $t$-th token in the sequence. In an $L$-layer transformer, the parameters associated with self-attention are given by $\{\mathbf{W}^{(l)}, \mathbf{P}^{(l)}, \mathbf{V}^{(l)}\}_{l=1}^{L}$ together with $\mathbf{W}_O$, where $\mathbf{W}^{(l)} \in \mathbb{R}^{|\mathcal{V}| \times |\mathcal{V}|}$ is the key–query matrix of layer $l$, $\mathbf{V}^{(l)} \in \mathbb{R}^{|\mathcal{V}| \times |\mathcal{V}|}$ is the value matrix, $\mathbf{P}^{(l)} \in \mathbb{R}^{T}$ is the learned relative positional encoding, and $\mathbf{W}_O \in \mathbb{R}^{|\mathcal{V}| \times |\mathcal{V}|}$ is the output matrix. The model with input $\mathbf{X}$ is defined as follows.

**Definition 3.1** (Attention-Based Transformer). *Given an input matrix* $\mathbf{X} \in \mathbb{R}^{T \times |\mathcal{V}|}$, *the L-layer attention-based transformer with parameters* $\Theta = \{\mathbf{W}^{(l)}, \mathbf{P}^{(l)}, \mathbf{V}^{(l)}\}_{l=1}^{L} \cup \{\mathbf{W}_O\}$ *is defined as*

$$\mathbf{F}_{\Theta}(\mathbf{X}) = \mathbf{h}^{(L)}\mathbf{W}_O, \tag{1}$$

*where* $\mathbf{h}^L$ *is defined by the recurrence relation, i.e.,*

$$\mathbf{h}^{(l)} = \mathbf{h}^{(l-1)} + \mathcal{S}(Mask(\mathbf{h}^{(l-1)}\mathbf{W}^{(l)}\mathbf{h}^{(l-1)\top} + DM(\mathbf{P}^{(l)})))\mathbf{h}^{(l-1)}\mathbf{V}^{(l)} \text{ and } \mathbf{h}^{(0)} = \mathbf{X}, \tag{2}$$

*where* $\mathcal{S}(\cdot)$ *represents the softmax function,* $DM(v)$ *maps the ith element of v to the* $(-i+1)$*th subdiagonal, and* $Mask(\cdot)$ *denotes the operator of attention mask.* This architecture is in line with Nichani et al. (2024), and recent work shows that self-attention–only models can match the performance of architectures with MLP layers (Wang et al., 2025).

### 3.3 TRAINING SETUP

**Learning objective.** To align with standard language modeling practice and ensure comparability with prior works (Huang et al., 2025; Nichani et al., 2024), we adopt the standard cross-entropy objective: given $N$ input matrices $\mathbf{X}_1, ..., \mathbf{X}_N$ with sequence length $T$ and corresponding output matrices $\mathbf{Y}_1, ..., \mathbf{Y}_N$, where $\mathbf{Y}_i \in \mathbb{R}^{T \times |\mathcal{V}|}$, the objective function is defined as

$$\mathcal{L}(\Theta) = \frac{-1}{NT} \sum_{i=1}^{N} \sum_{t=1}^{T} \log \mathcal{S}(\mathbf{F}_{\theta}(\mathbf{X}_i)^{[t]}) \mathbf{Y}_i^{[t]\top}, \tag{3}$$

where $\mathbf{M}^{[t]}$ denotes the $t$-th row of a matrix $\mathbf{M}$ and $\mathbf{Y}_i^{[t]}$ corresponds to the one-hot embedding for the $t+1$-th token of the sequence corresponding to $\mathbf{X}_i$.

**Gradient descent.** We analyze the evolution of the parameters under full-batch gradient descent with a constant learning rate $\eta$. Under gradient descent, the parameters are updated as follows:

$$\Theta(t) = \Theta(t-1) - \eta \nabla_{\Theta} \mathcal{L}(\Theta). \tag{4}$$

Due to the nonlinear complexities of the gradient, deriving an exact form for even one of the weight matrices after $t$ steps is challenging. We address this challenge by considering a leading-order approximation technique, allowing for a closed-form expression of the gradients and weights while yielding a tight approximation of the full gradient.

## 4 THEORETICAL ANALYSIS

In Section 4.1, we provide theorems demonstrating that the weights of attention-based transformers remain close to their gradient leading terms for $O(1/\eta)$ steps under both zero and Gaussian initializations. Then, Section 4.2 uncovers how three basis functions, which are crucial to express token associations and language structure, are encapsulated in those gradient leading terms, and how these three functions are compounded to shape the desiderata of the transformers' weight matrices.

### 4.1 MAIN THEOREMS

Under the setup described in Sec. 3, we obtain the following results for attention-based transformers.

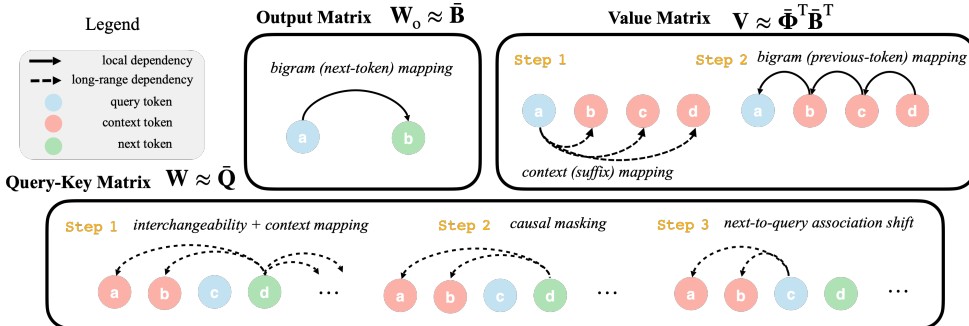

Figure 2: **Illustration of theoretical results.** We characterize weight matrices of the attention-only transformer as compositions of three basis functions: bigram mapping, interchangeability mapping, and context mappings. We illustrate how these mappings are composed across weight matrices to learn semantic associations between a given query token and its surrounding text.

**Theorem 4.1.** *(Informal) Given an attention-based transformer (Def. 3.1) under sufficiently small Gaussian initialization, with $L \leq \sqrt{T}/4$, after $s$ gradient descent steps with learning rate $\eta \geq \frac{1}{T}$, if $s \leq \eta^{-1} \min(\frac{5}{8\sqrt{T}}, \frac{1}{12L})$, then for all layers $l = 1, \ldots, L$,*

$$\left\| \mathbf{W}_O - s\eta\bar{\mathbf{B}} \right\|_F \leq 3s^2\eta^2, \tag{5}$$

$$\left\| \mathbf{V}^{(l)} - \binom{s}{2}\eta^2\bar{\mathbf{\Phi}}^\top\bar{\mathbf{B}}^\top \right\|_F \leq 12s^3\eta^3, \tag{6}$$

$$\left\| \mathbf{W}^{(l)} - \left(3\binom{s}{4} + 2\binom{s}{3}\right)\eta^4\bar{\mathbf{Q}} \right\|_F \leq 13s^5\eta^5 T, \tag{7}$$

$$\left\| \mathbf{P}^{(l)} - \left(3\binom{s}{4} + 2\binom{s}{3}\right)\eta^4\mathbf{\Delta} \right\|_F \leq 13s^5\eta^5 T, \tag{8}$$

*where $\| \cdot \|_F$ is the Frobenius norm, $\bar{\mathbf{B}}$ corresponds to a bigram statistic, $\bar{\mathbf{\Phi}}$ corresponds to a context co-occurrence statistic, $\bar{\mathbf{Q}}$ corresponds to a token-to-token correlation based on a composition of $\bar{\mathbf{B}}$ and $\bar{\mathbf{\Phi}}$, and $\mathbf{\Delta}$ corresponds to a relative position correlation based on the same feature as $\bar{\mathbf{Q}}$.*

The above Theorem shows that any finite-depth $L$-layer attention-based transformer (Def. 3.1) has the same characterization for its weights uniformly across all layers under a zero-initialization (Theorem D.10) and a small Gaussian initialization (Theorem 4.1), suggesting that all layers of the model capture common associative features from natural language as a starting point before evolving differently as training progresses (Figure 6). As seen in Figure 2, compositions of these features form the leading terms of the output matrix ($\bar{\mathbf{B}}$), value matrix ($\bar{\mathbf{\Phi}}^\top\bar{\mathbf{B}}^\top$), and query-key matrix ($\bar{\mathbf{Q}}$). We walk through these matrices in Section 4.2.1 and how they form the weights of the model in Section 4.2.2. The formal theorem and proofs are in Appendix D.

## 4.2 INTERPRETATION OF THEOREMS

In the previous section, we showed that the model parameters can be approximated by key corpus statistics $\bar{\mathbf{B}}, \bar{\mathbf{\Phi}}, \bar{\mathbf{Q}}$ and $\mathbf{\Delta}$. Now, we discuss the definitions of these statistics by first introducing *three basis functions* and explaining how *their composition characterizes the model's behavior*.

### 4.2.1 THREE BASIS FUNCTIONS SHAPING ASSOCIATIVE FEATURES

**(1) Bigram mapping $\bar{\mathbf{B}}$.** The $(i, j)$-th element in $\bar{\mathbf{B}}_{ij}$ corresponds to a correlation between token $\mathbf{e}_i$ and token $\mathbf{e}_j$ based on how likely $\mathbf{e}_i$ is to be directly followed by $\mathbf{e}_j$ as a bigram. More precisely,

$$\bar{B}_{ij} = \mathcal{P}_t(\mathbf{e}_i)\mathcal{P}_t(\mathbf{e}_j|\mathbf{e}_i) - \mathcal{P}_t(\mathbf{e}_i)/|\mathcal{V}|, \tag{9}$$

where $\mathcal{P}_t(\mathbf{e}_i)$ is the relative frequency of $\mathbf{e}_i$ over all tokens in the dataset $\mathbf{X}_1, ..., \mathbf{X}_N$ and $\mathcal{P}_t(\mathbf{e}_j|\mathbf{e}_i)$ is the relative frequency of $\mathbf{e}_j$ given that the previous token was $\mathbf{e}_i$. The product between $\mathcal{P}_t(\mathbf{e}_i)$ and $\mathcal{P}_t(\mathbf{e}_j|\mathbf{e}_i)$ forms an estimate of the likelihood of $\mathbf{e}_i$ followed by $\mathbf{e}_j$ appearing as a bigram and the second term $-\mathcal{P}_t(\mathbf{e}_i)/|\mathcal{V}|$ simply acts as a centering term such that each row sums to 0.

**(2) Interchangeability mapping $\Sigma_{\bar{\mathbf{B}}}$.** We study $\Sigma_{\bar{\mathbf{B}}} = \bar{\mathbf{B}}^\top \bar{\mathbf{B}}$, the correlation matrix of $\bar{\mathbf{B}}$, which captures correlations between pairs of tokens based on a frequency-weighted similarity of their previous-token distributions. From Eq. (9), neglecting the centering terms, the $(i, j)$-th element of $\Sigma_{\bar{\mathbf{B}}}$ can be represented as

$$\underbrace{\mathcal{P}_t(\mathbf{e}_i)\mathcal{P}_t(\mathbf{e}_j)}_{\text{Frequency weighting}} \sum_{k=1}^{|\mathcal{V}|} \underbrace{\mathcal{P}_t(\mathbf{e}_k^\leftarrow | \mathbf{e}_i)\mathcal{P}_t(\mathbf{e}_k^\leftarrow | \mathbf{e}_j)}_{\text{Previous token similarity}}. \tag{10}$$

In essence, Eq. (10) shows that $\Sigma_{\bar{\mathbf{B}}}$ captures a symmetric relationship between tokens based on how similar of a function or role they play across different contexts. Specifically, in Eq. (10), we can see that the corresponding row, which acts as a feature for token $\mathbf{e}_i$ captures its associations with **interchangeable** tokens captured by the previous token similarity factor and **frequent** tokens captured by the frequency weights. Similarities in previous token distributions are an indicator of functional similarities or interchangeability, as this captures structural patterns such as nouns being preceded by articles or adjectives and objects being preceded by common descriptors. This interchangeability map, $\Sigma_{\bar{\mathbf{B}}}$, acts a building block of characterizations for the weights $\mathbf{W}^{(l)}$ and $\mathbf{P}^{(l)}$ as illustrated in Figure 2. We depict a simple example of a word-wise correlation captured by $\Sigma_{\bar{\mathbf{B}}}$ in Figure 1.

**(3) Context mapping $\bar{\bar{\mathbf{\Phi}}}$.** The $(i, j)$-th element of $\bar{\bar{\mathbf{\Phi}}}$ corresponds to a correlation between token $\mathbf{e}_i$ and $\mathbf{e}_j$ based on how likely $\mathbf{e}_j$ is to appear as a prefix of $\mathbf{e}_i$. This can be written as

$$\frac{1}{T} \sum_{k=1}^{T} \frac{1}{k} \sum_{m=1}^{k} \mathcal{P}_t(\text{the } k+1 \text{ -th token is } \mathbf{e}_i, \text{the } m \text{ -th token is } \mathbf{e}_j) - \mu_j, \tag{11}$$

where $\mu_j$ centers the columns of $\bar{\bar{\mathbf{\Phi}}}$ to be 0. Considering each row as an embedding for a token $\mathbf{e}_i$, which represents an average of the tokens that appear in its context, i.e., smoothed context.

More precisely, the strength of the association from token $\mathbf{e}_i$ to $\mathbf{e}_j$ is determined by the average probability that $\mathbf{e}_j$ appears in the context of $\mathbf{e}_i$ over possible positions of $\mathbf{e}_i$ and $\mathbf{e}_j$. This matrix can be interpreted as assigning a representation to a token based on a summary of the possible contexts that token $\mathbf{e}_i$ appears in. This allows for learning associations between words that capture richer semantic relationships than bigram features. For example, we could expect to see correlations between animal and habitat, country and capital, or emotions and facial expressions (See Figure 3). This context mapping $\bar{\bar{\mathbf{\Phi}}}$ is a core building block of the gradients for the query-key attention $\mathbf{W}^{(l)}$ and value $\mathbf{V}^{(l)}$ matrices as shown in Figure 2.

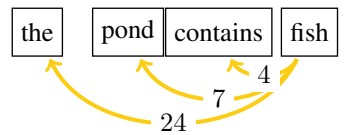

Figure 3: An example of $\bar{\bar{\mathbf{\Phi}}}$ with arrows pointing to prefix tokens for "fish" with context summary scores on edges. Larger values indicate the token appears more frequently in the context of "fish".

### 4.2.2 COMPOSITION OF BASIS FUNCTIONS FOR SEMANTIC ASSOCIATION

We now show how these three basis functions, bigram mapping $\bar{\mathbf{B}}$, interchangeability mapping $\Sigma_{\bar{\mathbf{B}}}$, and context mapping $\bar{\bar{\mathbf{\Phi}}}$, are compounded to characterize four classes of weight matrices of the transformer.

**(1) Output matrix $\mathbf{W}_O$.** As shown in Eq. (5), $\bar{\mathbf{B}}$ is the leading term of $\mathbf{W}_O$, and thus the mapping from embedding vectors to output predictions can be understood by examining the matrix product $\mathbf{e}_i\bar{\mathbf{B}}$ for a token embedding $\mathbf{e}_i$. The $j$-th element of the resulting output vector is $\bar{\mathbf{B}}_{ij}$, and each $\bar{\mathbf{B}}_{ij}$ includes a factor of $\mathcal{P}_t(\mathbf{e}_i)$. This implies that tokens are scored according to how frequently they occur in the average next-token distribution of $\mathbf{e}_i$, and explain how models at early stages effectively learn bigram-like patterns.

**(2) Value matrix $\mathbf{V}^{(l)}$.** The leading term of the value matrix $\mathbf{V}^{(l)}$ can be expressed as $\bar{\bar{\mathbf{\Phi}}}^\top \bar{\mathbf{B}}^\top$ as noted in Eq. (6), which acts as a composition of a context summary and bigram mapping. Because

$\bar{\mathbf{\Phi}}^{\top}$ captures longer-term dependencies and $\bar{\mathbf{B}}^{\top}$ captures only bigram statistics, the resulting embedding from $\mathbf{V}^{(1)}$ still endows the original token representations with semantic properties similar to those of $\bar{\mathbf{\Phi}}^{\top}$ as seen in Figure 2.

**(3) Attention matrix $\mathbf{W}^{(l)}$.** Theorem 4.1 characterizes the attention weight (a shared query-key matrix) as $\bar{\mathbf{Q}}$, which is constructed as a composition of $\mathbf{\Sigma}_{\bar{\mathbf{B}}}$, $\bar{\mathbf{\Phi}}$, the input matrix $\mathbf{X}_i$, the output matrix $\mathbf{Y}_i$, etc. We note that this compound feature captures a token-to-token correlation determined by how predictive one token is of the other's next-token distribution based on the context and interchangeability mappings. We walk through an overview of the construction of $\bar{\mathbf{Q}}$ in three steps (See Appendix A for details).

1. *Input-output matching scoring in context.* As a preliminary step, we first define a composed feature $\mathbf{\Sigma}_{\bar{\mathbf{B}}}\bar{\mathbf{\Phi}}$ by multiplying the interchangeability mapping $\mathbf{\Sigma}_{\bar{\mathbf{B}}}$ with the context mapping $\bar{\mathbf{\Phi}}$. This composition utilizes local interchangeability to map a token to a class of similar tokens and utilizes the context mapping to capture longer-range semantic correlations shared by the set of similar tokens. Using this feature, for each sample, we assign scores between each input and output token.

2. *Masking and centering.* The auto-regressive constraint is enforced by masking future tokens, keeping only scores from input tokens that precede the output token. Then, the resulting scores for each output token are centered and normalized based on its position.

3. *Next-to-query shift and averaging.* The scores between each input and output token are then shifted so that the same score is assigned instead to be between the input token and the token directly preceding the output token. Then, the scores are averaged across all samples.

**(4) Positional encoding $\mathbf{P}^{(l)}$.** The closed-form characterization $\mathbf{\Delta}$ of the positional encoding $\mathbf{P}^{(l)}$ follows a very similar composition to $\bar{\mathbf{Q}}$, with the main difference being that the correlations are mapped to *positional differences* rather than to the vocabulary-space differences (See Lemma D.1).

### 4.2.3 HOW THE WEIGHTS COOPERATE

To illustrate how the weights work together and provide further context on the role of each of the weights as functions, we consider the leading-term computation of a single-layer attention-based model. Dropping constant factors to focus on the interactions between features, the leading terms of the entire model computation can be written as

$$\left(\mathcal{S}\left(\text{Mask}\left(\mathbf{X}\bar{\mathbf{Q}}\mathbf{X}^{\top} + \text{DM}(\mathbf{\Delta})\right)\right)\mathbf{X}\bar{\mathbf{\Phi}}^{\top}\bar{\mathbf{B}}^{\top} + \mathbf{X}\right)\bar{\mathbf{B}}. \tag{12}$$

We can further decompose this into $\mathbf{X}\mathbf{W}_O$ and the computation from the self-attention block is:

$$\mathcal{S}\left(\text{Mask}\left(\mathbf{X}\bar{\mathbf{Q}}\mathbf{X}^{\top} + \text{DM}(\mathbf{\Delta})\right)\right)\mathbf{X}\bar{\mathbf{\Phi}}^{\top}\mathbf{\Sigma}_{\bar{\mathbf{B}}}. \tag{13}$$

$\bar{\mathbf{Q}}$ and $\mathbf{\Delta}$ capture correlations between two tokens or two positions based on how predictive the first token/position is of the next-token distribution of the second token/position according to $(\bar{\mathbf{\Phi}}^{\top}\mathbf{\Sigma}_{\bar{\mathbf{B}}})^{\top}$. Notice that the attended tokens are mapped to the output space by $\bar{\mathbf{\Phi}}^{\top}\mathbf{\Sigma}_{\bar{\mathbf{B}}}$, the same feature that determines the correlations for attention. As a result, the self-attention block effectively attends to tokens that, under the value and output matrix projection, lead to better next-token prediction. Thus, we find that while the residual stream $\mathbf{X}\mathbf{W}_O$ provides an average prediction of the next token, $\bar{\mathbf{Q}}$ enables the model to refine this prediction by selectively focusing on tokens most indicative of the next-token given its current parameters, those capturing corpus association statistics.

> **Implication.** By considering an end-to-end analysis of the model under simultaneous training of layers and by decomposing the weights, we obtain a clear interpretation of how different components collaborate to form semantic representations and can rigorously contextualize the function of each component in the full computation of attention-based transformers. While these features only yield small changes in the actual text output, they provide important insight into how the model's behavior develops during training. For example, if early training already associates *fish* with *pond* (as in Figure 3), we expect such relationships to be a useful anchor for later training, allowing the model to complete more complex sentences, e.g., "A pond in the garden was filled with colorful fish that sparkled in the sunlight", coherently with learned semantic associations.

## 5 EXPERIMENTS

### 5.1 3-LAYER ATTENTION-BASED TRANSFORMER

We begin with an experimental setting designed to closely mirror our theory, enabling direct verification of results and analysis of the semantic relationships embedded in the learned weights. For clearer interpretability, we use the TinyStories dataset (Eldan & Li, 2023), truncated to the 3,000 most frequently occurring words, which also defines the model's vocabulary. A 3-layer self-attention model defined in Definition 3.1 is then trained with sequence length $T = 200$.

Table 1: Minimum cosine similarities between theoretical and actually learned weights across all epochs. Results from a 3-layer attention-based model trained on TinyStories (small $\eta$).

| Weights | Min. Cosine |
|---|---|
| Attention | 0.999496 |
| Value | 0.999169 |
| Output | 0.998486 |

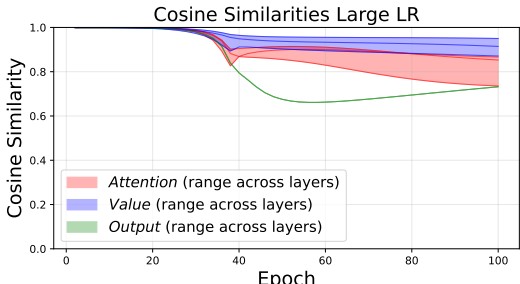

Figure 4: **Cosine similarity between theoretical and learned weights**. Results from a 3-layer transformer model trained on TinyStories.

**Verification of theory.** To verify Theorem 4.1, we measure the cosine similarity between the learned weights and their corresponding leading terms at checkpoints over the first 100 epochs of SGD using a batch size of 2048 for computational tractability with a learning rate of 0.005. We also consider the cosine similarity between the learned weights and their leading terms when using a larger learning rate of 0.05 to understand how features evolve at later stages with respect to the leading term gradients. We provide results for both settings in Table 1 and Figure 4. The results show that *the learned weights maintain strong agreement with the theoretical predictions: even after 30 epochs, all weights achieve a cosine similarity of at least 0.9.* Moreover, all parameter matrices have a cosine similarity above 0.7, even after 100 epochs, where the loss had dropped from 8.00 to 5.35. These findings suggest that the features predicted by the theorem not only characterize the model dynamics during the early stage, but also remain informative well beyond it. We provide results for a BPE tokenization and for a causal analysis in Appendix B, and we elaborate experimental details for the TinyStories experiments in Appendix C.

| the | red | to | | they | happy | wanted | | fish | flower | birds |
|---|---|---|---|---|---|---|---|---|---|---|
| park | ball | the | | she | happy | saw | | fish | beautiful | bird |
| little | car | play | | they | sad | had | | big | yellow | tree |
| bird | dress | go | | he | excited | wanted | | small | butterfly | up |
| ball | balloon | be | | it | scared | asked | | pond | hose | park |
| dog | truck | help | | one | proud | went | | lake | garden | nest |
| big | blocks | see | | lily | angry | loved | | water | bloom | flowers |
| tree | apples | her | | timmy | nice | ran | | catch | pretty | tweety |
| man | shirt | make | | tom | curious | looked | | sea | daisy | sky |
| box | hat | do | | her | surprised | took | | boat | field | flew |
| (a) Examples for $\bar{\mathbf{B}}$ | | | | (b) Examples for $\mathbf{\Sigma}_{\bar{\mathbf{B}}}$ | | | | (c) Examples for $\bar{\mathbf{\Phi}}$ | | |

Figure 5: Selected tokens from the top 30 correlated tokens under different basis features from TinyStories. The characterized features actually capture both grammatical and semantic structures.

**Semantic structure.** To validate our interpretation of associative features, we collect for each token the top 30 most correlated tokens under each of the basis functions: the bigram mapping ($\bar{\mathbf{B}}$), interchangeability mapping ($\mathbf{\Sigma}_{\bar{\mathbf{B}}}$), and context matrix ($\bar{\mathbf{\Phi}}$), constructed from the TinyStories corpus. We provide examples of tokens where the expected semantic relationships can be observed in Figure 5. Under $\bar{\mathbf{B}}$, we see that the word "red" is correlated with common objects such as "truck"

that would be described by the word "red". Under $\bar{\bar{\mathbf{\Phi}}}$, we can see that the word "fish" is correlated with common settings where fish would appear, such as "pond" or "lake".

## 5.2 TRANSFORMERS IN PRACTICE

**Setup.** To evaluate how well our theoretical results extend to practical LLMs, we analyze token relationships learned from OpenWebText (Gokaslan et al., 2019), a real-world large-scale dataset with text from millions of webpages, in Pythia-1.4B (Biderman et al., 2023) and compare them with our theoretical predictions, examining how these relationships evolve across layers on datasets and models reflecting real-world complexities. We choose the Pythia model family, as they are open-sourced and uniquely provide access to intermediate checkpoints, enabling fine-grained analysis of training dynamics and interpretability (Marks et al., 2024; Gallego-Feliciano et al., 2025). Unlike our theoretical setting, Pythia includes additional components such as MLP and multi-head attention, making it impossible to directly read off average token correlations from the weights. In order to interpret the layer-wise representations in terms of token-token correlations, we perform the analysis through the following steps:

1. We pass in each token $\mathbf{e}_i$ as the input to the transformer.

2. For each token and from each layer $l$, we collect the following embeddings: the input to layer $l$ $\mathbf{h}_{i,l,pre}$, the output of the $l$-th layer $\mathbf{h}_{i,l,post}$, and the output of the $l$-th layer without the MLP component $\mathbf{h}_{i,l,attn}$.[1]

3. The embeddings $\mathbf{h}_{i,l,pre}$ form the rows of $\mathbf{E}_{l,pre} \in \mathbb{R}^{|\mathcal{V}| \times d}$ which represents a mapping from the input embeddings of layer $l$ to tokens. Similarly, the embeddings $\mathbf{h}_{i,l,post}$ and $\mathbf{h}_{i,l,attn}$ form the rows of $\mathbf{E}_{l,post} \in \mathbb{R}^{|\mathcal{V}| \times d}$ and $\mathbf{E}_{l,attn} \in \mathbb{R}^{|\mathcal{V}| \times d}$ respectively.

**Attention correlations.** To analyze the correlations captured by the attention weights at each layer, we compute the product of the key and query mappings for each head and average these products, which we will call $\mathbf{A}_{l,emb} \in \mathbb{R}^{d \times d}$. We then multiply the mapping $\mathbf{E}_{l,pre}$ on both sides of $\mathbf{A}_{l,emb}$ to convert the average attention mapping into a token-basis attention weight matrix $\mathbf{A}_{l,tok}$. Finally, we consider token correlations captured by $\mathbf{A}_{l,tok}$ by using its covariance matrix, which we compare with the covariance matrix of $\bar{\mathbf{Q}}$, the leading-order attention mapping term from our theorem.

**Embedding correlations.** To analyze the correlations captured by the value mapping and the MLP, we consider the token-token correlations captured by the output of each layer. Utilizing the covariance matrix of $\mathbf{E}_{l,post}$ allows for direct comparison with the covariance matrix of the leading value matrix term $\bar{\mathbf{\Phi}}^{\top}\bar{\mathbf{B}}^{\top}$, since the matrices themselves have different dimensions. Furthermore, this enables us to control for shifts in the embedding space.

**Comparison methodology.** We compute the leading term matrices using 100K samples from OpenWebText. To control for differences in model architecture, we normalize each row of the leading term weights to have unit norm. Then, we compute cosine similarities between the corresponding covariance matrices across layers and across checkpoints. We perform the same analysis on the FineWeb (Penedo et al., 2024) dataset and provide results in Appendix B. More details on the experimental setup are in Appendix C.



Figure 6: Cosine similarity between covariance matrices for Pythia-1.4B attention weights and embeddings and the corresponding leading term features based on OpenWebText.

---

[1]We remind the reader of each layer's structure in the Pythia model. The input is normalized and then passed into the attention block and the MLP block in parallel. Then the outputs of each are added to the original input.

**Results.**   We provide a visualization of results in Figure 6, where we can see that, at the early stage of training, there is very strong agreement between the Pythia embeddings and our leading-term features. We can see that for the embedding mapping, *the token representations strongly match our theoretical analysis across all layers, and similarly for the attention weights*, excluding only the first layer. We can see that as the model continues training, the weights gradually drift from fixed associative features to represent richer knowledge beyond association, starting with the earlier layers. However, it still maintains these features to a large extent for relatively longer steps. This suggests that our analysis on attention-based models generalizes with the addition of multi-head attention or MLP and acts as a starting point for a finer-grained analysis of full training dynamics.

**MLP ablation.**   We perform an ablation at each layer by performing the embedding correlation analysis using $\mathbf{E}_{l,attn}$, which is based on only the output of the attention block and excludes the MLP component. The results for this analysis can be seen in the middle plot of Figure 6. We can see that the correlations captured by embeddings with and without the MLP are similar except at the first layer. This suggests that at the first layer, the MLP maps tokens to embeddings with structures similar to that of the leading-order value matrix term and maintains a similar structure at later layers. Based on these initial results, one possible hypothesis is that the MLP at early stages functions similarly to the leading-term value mapping.

**Individual attention heads.**   In order to capture a fine-grained understanding of the attention block, we perform the analysis on attention correlations using individual attention heads. We perform this analysis at an early (Layer 2), middle (Layer 13), and late layer (Layer 24) to also understand how heads may evolve differently at different stages of the model. In Figure 7, we find that different layers evolve differently with respect to the gradient leading-term for attention mappings. The earlier layers learn the leading-term features at a slower rate, as seen by the high similarity (red) appearing at later steps, especially for layer 2. We can also see that layer 13 exhibits faster specialization of attention heads than the other layers, as seen by the high variance in each column at later steps for layer 13. This provides insight into the rate of specialization of attention heads and suggests that intermediate layers are where specialization initially occurs.

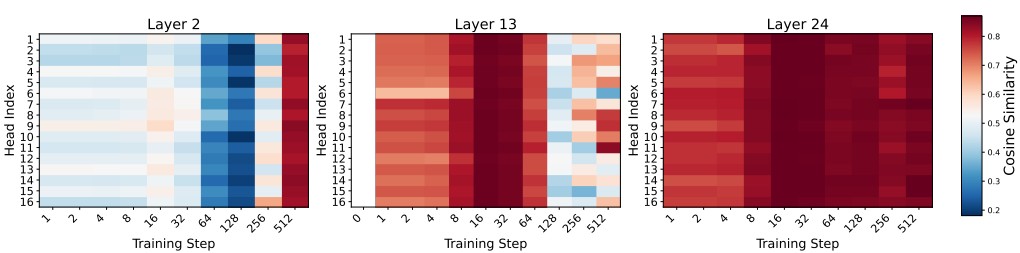

Figure 7: Cosine similarity between covariance matrices for Pythia-1.4B individual attention head weights and the corresponding leading term features based on OpenWebText.

## 6   CONCLUSION

We present new theoretical results on the emergence of semantic associations in self-attention models learned from a natural language dataset. Our gradient leading term analysis for each model weight illuminates how the core basis functions that shape the associative features, i.e., bigram mapping, interchangeability mapping, and context mapping, develop from the training corpus. We show that transformer weights have closed-form expressions as compositions of those basis functions to represent semantic associations across natural language tokens. The extensive analyses on the weight matrices' characterizations grounded by empirical supports from toy transformers and real-world LLMs contribute to the theoretical foundations of representation learning in transformers while also opening pathways for interpretability research: discovering common factors that allow weight matrices across components to be decomposed into simple functions of those shared factors; leveraging theory to formulate broad hypotheses about how concepts arise in models, extending beyond individual mechanisms or specific behaviors to complex characteristics.

ETHICS STATEMENT

We provide a novel theorem that characterizes the roles of weights in the transformer model, which is a de facto standard building block of modern LLMs. We try to uphold high standards of scientific excellence by making minimal assumptions for theoretical analysis while providing practical implications on mechanistic interpretability. The new insights on emerging features we presented contribute to a better understanding and diagnosis of the representation learning of transformers, which makes a big step towards transparent and reliable AI. As our study considers a setup of training from scratch on public datasets, there is no direct privacy issue and harm. The authors also acknowledge and respect the ethics of confidentiality and fairness, and confirmed that there are no identified violations of them.

REPRODUCIBILITY STATEMENT

All the theoretical analyses in this work are accompanied by full proofs with detailed step-by-step explanations (in the Appendix) for verification, reproduction, and reuse. We elaborate on the details of our setup in the main body of the paper for all empirical validations, and the code is available here. In addition, our choice of models pursues maximum reproducibility and accessibility, given its simple and fully open-sourced configurations.

ACKNOWLEDGMENTS

The authors would like to thank James Oldfield, Wendi Li, and Jimmy Di for their valuable feedback. The work is supported in part by the AFOSR Young Investigator Program under award number FA9550-23-1-0184, National Science Foundation under awards IIS-2237037 and IIS-2331669, Office of Naval Research under grant number N00014-23- 1-2643, Schmidt Sciences Foundation, Open Philanthropy, Alfred P. Sloan Fellowship, and gifts from Google and Amazon. Zhen Fang was funded by the Australian Government through the Australian Research Council (ARC) under grant number DE250100363. Shawn Im is also supported by the National Science Foundation Graduate Research Fellowship Program under Grant No. 2137424. Any opinions, findings, and conclusions or recommendations expressed in this material are those of the author(s) and do not necessarily reflect the views of the National Science Foundation. Support was also provided by the Graduate School and the Office of the Vice Chancellor for Research at the University of Wisconsin-Madison with funding from the Wisconsin Alumni Research Foundation.

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

## A    DETAILED DESCRIPTION ON WEIGHT CHARACTERIZATION

The token-to-token correlation captured by $\bar{\mathbf{Q}}$ is determined by how strongly correlated one token is with the other's next-token distribution. These correlations are captured by $Q_i$ where each element $Q_{i_{jk}}$ of $Q_i$ measures for $X_i$, the correlation between the token at position $j$ and the token at position $k + 1$. This correlation between the token at position $j$ and position $k + 1$ gets mapped back to a correlation between the tokens at positions $j$ and $k$ through $\mathbf{X}_i^\top \mathbf{Q}_i \mathbf{X}_i$. let $\mathbf{Q}_i$ given in Eq. (16) be the per-example correlation matrix computed from input–output token pairs in $i$th input,

$$\bar{\mathbf{Q}} = \frac{1}{NT} \sum_{i=1}^{N} \mathbf{X}_i^\top \mathbf{Q}_i \mathbf{X}_i. \tag{14}$$

We walk through an overview of the construction of $\mathbf{Q}_i$ in four steps and provide the detailed computation in Appendix A.

**Feature composition.** As a preliminary step, we first define a composed feature $\boldsymbol{\Sigma}_{\bar{\mathbf{B}}}\bar{\boldsymbol{\Phi}}$ by multiplying the interchangeability mapping $\boldsymbol{\Sigma}_{\bar{\mathbf{B}}}$ with the context mapping $\bar{\boldsymbol{\Phi}}$. Each entry corresponds to the average product of path weights from token $\mathbf{e}_i$ to $\mathbf{e}_j$ with one step on $\boldsymbol{\Sigma}_{\bar{\mathbf{B}}}$ and one step on $\bar{\boldsymbol{\Phi}}$. This composition utilizes local interchangeability to map a token to its more general functional class and utilizes the context-summary to capture longer-range semantic correlations shared by tokens in the functional class. We will refer to the resulting feature as the composed feature for simplicity in the remaining steps.

**Scoring input–output pairs.** For each input $\mathbf{X}_i$ and its corresponding output $\mathbf{Y}_i$, we utilize the composed feature, $\boldsymbol{\Sigma}_{\bar{\mathbf{B}}}\bar{\boldsymbol{\Phi}}$, to compute correlation scores between input and output tokens as seen in Figure 2.

$$(\mathbf{Y}_i - \mathbf{U}_O)\boldsymbol{\Sigma}_{\bar{\mathbf{B}}}\bar{\boldsymbol{\Phi}}\mathbf{X}_i^\top, \tag{15}$$

where $\mathbf{U}_O$ is a baseline matrix with all elements set to $1/|\mathcal{V}|$. This assigns a correlation score to each input–output token pair according the composed feature.

**Masking and centering.** The auto-regressive constraint is enforced by masking future tokens, keeping only scores from input tokens that precede the output token. The resulting scores for each output token are centered and normalized based on its position. The matrix is then centered so that the scores for each output token sum to zero, yielding the per-example matrix $\mathbf{Q}_i$.

$$\mathbf{Q}_i = \text{ein}_{tjk,\,tk \to tj}\left(\mathbf{J}_i, (\mathbf{Y}_i - \mathbf{U}_O)\boldsymbol{\Sigma}_{\bar{\mathbf{B}}}\bar{\boldsymbol{\Phi}}\mathbf{X}_i^\top\right), \tag{16}$$

where $\mathbf{J}_i$ is the masking operator and ein denotes an Einstein summation.

**Next to Query Mapping.** Lastly, the scores between each input and output token are then mapped to be the correlation between the input token and the token preceding the output token. In this way, the model learns to attend to the input token when it expects the next token to be the output token.

**Aggregation across the dataset.** Finally, we map per-example correlations back to the vocabulary space and average over all $N$ inputs and $T$ tokens per input:

$$\bar{\mathbf{Q}} = \frac{1}{NT} \sum_{i=1}^{N} \mathbf{X}_i^\top \mathbf{Q}_i \mathbf{X}_i. \tag{17}$$

In this way, each token is associated with the average correlations to other tokens across the dataset.

## B    ADDITIONAL EXPERIMENTS

**BPE tokenization.**   We train a 3-layer attention-based model on TinyStories as in Section 5.1 using a BPE tokenization with vocabulary size of 10,000. We train the model for 10 epochs with a learning rate of 0.005 and measure the cosine similarity between the theoretical and actual weights. We report the minimum over the 10 epochs in Table 2.

| Weights | Min. Cosine |
|---|---|
| Attention | 0.999914 |
| Value | 0.998800 |
| Output | 0.997891 |

Table 2: Minimum cosine similarities between theoretical and actually learned weights across all epochs. Results from a 3-layer attention-based model trained on TinyStories and with a BPE tokenization.

**Causal intervention.** We aim to understand how the model output changes when removing the leading terms from each of the weights. We perform this analysis on the 3-layer attention-based transformers trained on TinyStories with a learning rate of 0.05. Unlike most causal intervention settings, the features considered have a general function rather than a specific function applicable to a narrower setting, and therefore, we expect removing the leading terms to result in performance degradation across the dataset. As a result, we choose to focus on the extent to which the output distribution changes when the leading term component is removed for each weight matrix. For each weight matrix, we remove the projection of the weight matrix onto its corresponding leading term. After removing this projection, we compute the loss of the resulting model on the dataset. We provide the results of this intervention in Table 3. We can see that the output layer has the largest effect on the loss, while the attention weights have the least. This behavior is predicted by the theory as the output layer has the largest order update, while the attention weights have the smallest order updates.

| Weights | Loss |
|---|---|
| Original | 5.349 |
| Attention Layer 0 | 5.350 |
| Attention Layer 1 | 5.352 |
| Attention Layer 2 | 5.361 |
| Value Layer 0 | 6.192 |
| Value Layer 1 | 6.526 |
| Value Layer 2 | 6.520 |
| Output | 8.287 |

Table 3: Loss of the attention-based model on TinyStories after the leading term component from each weight matrix is removed. The first row corresponds to the original model.

**Validation on additional dataset.** We perform the analysis in Section 5.2 on the token-token correlations captured by embeddings in Pythia-1.4B except instead of using OpenWebText, we use FineWeb (Penedo et al., 2024). We provide the results in Figure 8 where we see very similar results as with OpenWebText.

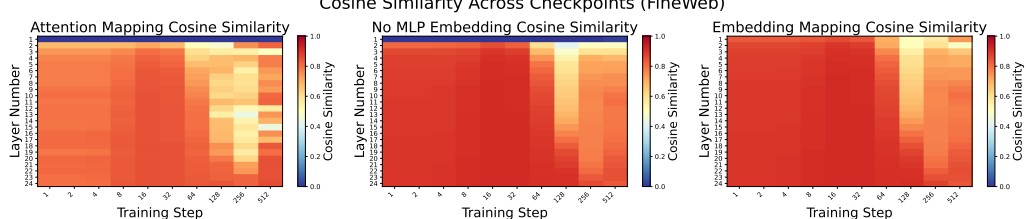

Figure 8: Cosine similarity between covariance matrices for Pythia-1.4B attention weights and embeddings and the corresponding leading term features based on FineWeb.

## C    Experimental Details

**TinyStories Experiments**   We collect the vocabulary from TinyStories treating each word, punctuation mark, or number as a token and use the 3000 most common tokens. We then filter out samples that include tokens outside of the set of 3000. For training, we use 65536 of the filtered samples with sequence length at least 201 and truncate all sequences to 201 tokens for training and computing theoretical leading terms. For the BPE tokenization, we tokenize the dataset using a vocabulary size of 10,000, and for training, we use samples with sequence length at least 201 and truncate all sequences to 201 tokens for training and computing theoretical leading terms. We compute the theoretical matrices using the first batch.

**Pythia Experiments**   We use the first 100k samples of OpenWebText/FineWeb with length at least 512 characters to perform the analysis.

We utilize 4 A100 GPUs with 80GB of memory. These experiments can be performed with less compute by reducing batch size or sequence length.

## D    Proofs

$\|\cdot\|$ will be the operator norm unless denoted otherwise.

### D.1    Proof of 1-Layer Theorem

**Lemma D.1** (General Gradient Form). *Under the setting described, we have that*

$$\frac{\partial \mathcal{L}}{\partial W_O} = \frac{-1}{NT} \sum_{i=1}^{N} h_i^{(1)\top} R_i \tag{18}$$

$$\frac{\partial \mathcal{L}}{\partial V^{(1)}} = \frac{-1}{NT} \sum_{i=1}^{N} X_i^\top A_i^{(1)\top} R_i W_O^\top \tag{19}$$

$$\frac{\partial \mathcal{L}}{\partial W^{(1)}} = \frac{-1}{NT} \sum_{i=1}^{N} X_i^\top ein_{tjk,tk \to tj}(J_i, (R_i W_O^\top V^{(1)\top} X_i^\top)) X_i \tag{20}$$

$$\frac{\partial \mathcal{L}}{\partial P^{(1)}} = \frac{-1}{NT} ein_{tjk,jk \to t} \left( D, \sum_{i=1}^{N} ein_{tjk,tk \to tj}(J_i, (R_i W_O^\top V^{(1)\top} X_i^\top)) \right) \tag{21}$$

*where $A_i^{(1)} = \mathcal{S}(Mask(X_i W^{(1)} X_i^\top + \mathrm{DM}(P^{(1)})))$, $R_i = Y_i - \mathcal{S}(F_\theta(X_i))$, $J_i \in \mathbb{R}^{T \times T \times T}$ with $J_{i,t} = Diag(A_i^{(1)[t]}) - A_i^{(1)[t]\top} A_i^{(1)[t]}$ being the Jacobian of the softmax function for the tth token in the sequence, $D \in \mathbb{R}^{T \times T \times T}$ with $D_t$ being a matrix with ones along the $(-t+1)$th sub-diagonal and zeros elsewhere, and $ein$ is used to denote an Einstein summation.*

*Proof.* We start by considering the derivative of the loss with respect to $F_\theta(X_i)^{[t]}$ which is

$$Y_i^{[t]} - \mathcal{S}(F_\theta(X_i)^{[t]}) \tag{22}$$

and derivative of $F_\theta(X_i)^{[t]}$ with respect to $W_O$ is $h_i^{(1)[t]}$. Then it follows that

$$\frac{\partial \mathcal{L}}{\partial W_O} = \frac{-1}{NT} \sum_{i=1}^{N} h_i^{(1)\top} R_i \tag{23}$$

Now, we consider the gradient with respect to $V^{(1)}$ using the chain rule which gives

$$\frac{\partial \mathcal{L}}{\partial V^{(1)}} = \frac{-1}{NT} \sum_{i=1}^{N} X_i^\top A_i^{(1)\top} R_i W_O^\top \tag{24}$$

Now, we consider the gradient with respect to $A_i^{(1)}$ as an intermediate step towards the gradient with respect to $W^{(1)}, P^{(1)}$. Using the chain rule as before, we have

$$\frac{\partial \mathcal{L}}{\partial A_i^{(1)}} = \frac{-1}{NT} R_i W_O^\top V^{(1)\top} X_i^\top. \tag{25}$$

Letting $B_i^{(1)} = X_i W^{(1)} X_i^\top + \mathrm{DM}(P^{(1)})$, we have that the derivative of $A_i^{(1)[t]}$ with respect to $B_i^{(1)[t]}$ is

$$J_{i,t} = \mathrm{Diag}(A_i^{(1)[t]}) - A_i^{(1)[t]\top} A_i^{(1)[t]} \tag{26}$$

Then, in order to get the gradient of the loss with respect to $B_i^{(1)}$, we need to consider the contribution from each $t$ resulting in the Einstein summation

$$\frac{\partial \mathcal{L}}{\partial B_i^{(1)}} = \frac{-1}{NT} ein_{tjk,tk \to tj}(J_i, R_i W_O^\top V^{(1)\top} X_i^\top) \tag{27}$$

From the chain rule, we can derive the gradient with respect to both $W^{(1)}$ and $P^{(1)}$.

$$\frac{\partial \mathcal{L}}{\partial W^{(1)}} = \frac{-1}{NT} \sum_{i=1}^{N} X_i^\top ein_{tjk,tk \to tj}(J_i, R_i W_O^\top V^{(1)\top} X_i^\top) X_i \tag{28}$$

$$\frac{\partial \mathcal{L}}{\partial P^{(1)}} = \frac{-1}{NT} ein_{tjk,jk \to t}\left(D, \sum_{i=1}^{N} ein_{tjk,tk \to tj}(J_i, R_i W_O^\top V^{(1)\top} X_i^\top)\right) \tag{29}$$

where $D_t$ has ones along the $(-t+1)$th sub-diagonal and zeros elsewhere. This completes the proof. $\qquad\square$

**Lemma D.2** (Local softmax Jacobian bound)**.** *Let $\mathcal{S}$ denote the softmax map. Then for every $z \in \mathbb{R}^d$,*

$$\|\nabla \mathcal{S}(z)\| \leq \max_{1 \leq j \leq d} \mathcal{S}(z)_j \leq \frac{\exp(\mathrm{range}(z))}{d},$$

*where*

$$\mathrm{range}(z) := \max_j z_j - \min_j z_j.$$

*For any $z$ such that $\mathrm{range}(z) \leq r$*

$$\left\|\mathcal{S}(z) - \frac{1}{d}\mathbf{1}\right\| \leq \frac{e^r}{d}\|z\|.$$

*Proof.* Let $p = \mathcal{S}(z)$. The Jacobian of softmax is

$$\nabla \mathcal{S}(z) = \mathrm{Diag}(p) - pp^\top.$$

For any $x \in \mathbb{R}^d$,

$$x^\top \nabla \mathcal{S}(z) x = \sum_{j=1}^{d} p_j x_j^2 - \left(\sum_{j=1}^{d} p_j x_j\right)^2.$$

The second term is nonnegative, so

$$x^\top \nabla \mathcal{S}(z) x \leq \sum_{j=1}^{d} p_j x_j^2 \leq \left(\max_j p_j\right) \|x\|_2^2.$$

Since $\nabla \mathcal{S}(z)$ is symmetric positive semidefinite, this implies

$$\|\nabla \mathcal{S}(z)\|_{2 \to 2} \leq \max_j p_j.$$

It remains to bound $\max_j p_j$. Let $M = \max_j z_j$ and $m = \min_j z_j$. Then for every $j$,

$$p_j = \frac{e^{z_j}}{\sum_{k=1}^{d} e^{z_k}} \leq \frac{e^M}{de^m} = \frac{e^{M-m}}{d} = \frac{e^{\mathrm{range}(z)}}{d}.$$

Therefore

$$\|\nabla\mathcal{S}(z)\|_{2\to2} \le \frac{e^{\mathrm{range}(z)}}{d}.$$

Now let $\gamma(\tau) = z' + \tau(z - z')$ for $\tau \in [0, 1]$. By the fundamental theorem of calculus,

$$\mathcal{S}(z) - \mathcal{S}(z') = \int_0^1 \nabla\mathcal{S}(\gamma(\tau))(z - z') \, d\tau.$$

If $\mathrm{range}(\gamma(\tau)) \le r$ for all $\tau \in [0, 1]$, then

$$\|\mathcal{S}(z) - \mathcal{S}(z')\|_2 \le \int_0^1 \|\nabla\mathcal{S}(\gamma(\tau))\|_{2\to2} \, d\tau \, \|z - z'\|_2 \le \frac{e^r}{d}\|z - z'\|_2.$$

Taking $z' = 0$ gives the final claim, since $\mathcal{S}(0) = \frac{1}{d}\mathbf{1}$ and

$$\mathrm{range}(\tau z) = \tau \, \mathrm{range}(z) \le \mathrm{range}(z).$$

$\square$

**Corollary D.3** (Local softmax bound from an $\ell_\infty$ bound). *Let $\mathcal{S} : \mathbb{R}^d \to \Delta^{d-1}$ denote the softmax map. If $\|z\|_\infty \le R$, then*

$$\left\|\mathcal{S}(z) - \frac{1}{d}\mathbf{1}\right\|_2 \le \frac{e^{2R}}{d}\|z\|_2.$$

*In particular, if $\|z\|_\infty \le \frac{1}{2}$, then*

$$\left\|\mathcal{S}(z) - \frac{1}{d}\mathbf{1}\right\|_2 \le \frac{e}{d}\|z\|_2.$$

*Proof.* If $\|z\|_\infty \le R$, then

$$\mathrm{range}(z) = \max_j z_j - \min_j z_j \le 2\|z\|_\infty \le 2R.$$

Applying Lemma D.2 with $r = 2R$ gives

$$\left\|\mathcal{S}(z) - \frac{1}{d}\mathbf{1}\right\|_2 \le \frac{e^{2R}}{d}\|z\|_2.$$

The final claim follows by taking $R = \frac{1}{2}$. $\square$

**Lemma D.4** (First Gradient Step). *Under the setting described, after one gradient step, we have that*

$$W_O = \eta(\bar{B}) \tag{30}$$

$$W^{(1)}, V^{(1)}, P^{(1)} = \mathbf{0} \tag{31}$$

*where $\bar{B}$ a $|V| \times |V|$ matrix where the $j$th row is the average next-token distribution of the $j$th token in the vocabulary weighted by the relative frequency of token $j$ across the dataset and centered to have the row sum be 0.*

*Proof.* From Lemma D.1, as the parameters are initially zero, we can see that $W^{(1)}, V^{(1)}, P^{(1)}$ all have gradients of zero and therefore remain as $\mathbf{0}$. For $W_O$, as the value matrix is initially zero, $h_i^{(1)} = X_i$ and as $W_O$ is zero, the output distribution for every token is the uniform distribution. Let $U_O \in \mathbb{R}^{T \times |V|}$ represent the resulting output with each element $1/|V|$. Then, we have that

$$\frac{\partial\mathcal{L}}{\partial W_O} = \frac{-1}{NT}\sum_{i=1}^N X_i^\top(Y_i - U_O) \tag{32}$$

We consider the sum of each of the terms $X_i^\top Y_i$ and $X_i^\top U_O$. First, we consider the sum of $X_i^\top Y_i$. The $j$th row of $X_i^\top Y_i$ is a $|V|$-dimensional vector with each element being the number of times the corresponding token appears after each occurrence of the $j$th token in the vocabulary in $X_i$. Then,

summing over all $i$ and dividing by $NT$ results in each row mapping to the average next-token distribution weighted by the frequency of the token corresponding to the row. We can write this as

$$B = \frac{1}{NT} \sum_{i=1}^{N} X_i^\top Y_i = \begin{bmatrix} \alpha_1 P_1 \\ \alpha_2 P_2 \\ \vdots \\ \alpha_{|V|} P_{|V|} \end{bmatrix} \tag{33}$$

where $\alpha_j$ is the relative frequency of the $j$th token in the dataset and $P_j$ is the average next-token distribution for token $j$. For the sum $X_i^\top U_O$ divided by $NT$, we simply get that every row is $\alpha_j$ times the uniform distribution over the vocabulary, and we will denote this matrix by $U$. Then, we have that

$$\frac{\partial \mathcal{L}}{\partial W_O} = -(B - U) \tag{34}$$

and therefore after the first step,

$$W_O = \eta(B - U) \tag{35}$$

Then, as $\bar{B} = B - U$, this completes the proof. $\qquad\square$

**Lemma D.5** (Second Gradient Step). *Under the setting described, and with $\eta \leq 1$, $|V| \geq 8$, after two gradient steps, we have that*

$$\left\| W_O - 2\eta\bar{B} \right\|_F \leq \frac{\eta^2}{\sqrt{|V|}} \tag{36}$$

$$\left\| V^{(1)} - \eta^2 \bar{\Phi}^\top \bar{B}^\top \right\|_F \leq \frac{2\eta^3}{\sqrt{|V|}} \tag{37}$$

$$W^{(1)}, P^{(1)} = \mathbf{0} \tag{38}$$

*where $\bar{B}$ is as defined in the previous lemma and $\bar{\Phi}$ is given by*

$$\bar{\Phi}_{jk} = \mathcal{P}(e_k \to e_j) - \mu_{\Phi,k} \tag{39}$$

*where $\mathcal{P}(e_k \to e_j)$ corresponds to the empirical probability that $e_j$ is the current token and $e_k$ is in its prefix and $\mu_{\Phi,k}$ is the value that sets each column sum to 0.*

*Proof.* First, as $V^{(1)}$ remains at zero after the first step, we have that the gradients for $W^{(1)}, P^{(1)}$ are zero and therefore, they remain at zero after the second step. We now consider the forward pass after the first gradient step. As the value matrix remains as zero, we have that

$$F_\theta(X_i) = \eta X_i \bar{B} \tag{40}$$

For each row $t$, let $f_{i,t} = F_\theta(X_i)^{[t,:]}$. Since $F_\theta(X_i) = \eta X_i \bar{B}$, each row $f_{i,t}$ is equal to $\eta$ times a row of $\bar{B}$. Since each row of $\bar{B}$ is a frequency-weighted centered probability vector, it has a range of at most 1. Hence

$$\text{range}(f_{i,t}) \leq \eta \leq 1.$$

By Lemma D.2, applied rowwise,

$$\left\| \mathcal{S}(f_{i,t}) - \frac{1}{|V|}\mathbf{1} \right\| \leq \frac{e}{|V|} \|f_{i,t}\|.$$

Summing over rows gives

$$\|\mathcal{S}(F_\theta(X_i)) - U_O\|_F \leq \frac{e}{|V|} \|F_\theta(X_i)\|_F = \frac{e\eta}{|V|} \left\| X_i \bar{B} \right\|_F \leq \frac{e\eta\sqrt{T}}{|V|}. \tag{41}$$

Then, as $|V| \geq 8$, we have that

$$\|R_i - (Y_i - U_O)\|_F \leq \frac{\eta\sqrt{T}}{\sqrt{|V|}} \tag{42}$$

and by Lemma D.1 and that $\|X_i\| \le \sqrt{T}$, we have

$$\left\| \frac{\partial \mathcal{L}}{\partial W_O} + \bar{B} \right\|_F \le \frac{1}{NT} \sum_{i=1}^{N} \frac{\eta T}{\sqrt{|V|}} = \frac{\eta}{\sqrt{|V|}} \tag{43}$$

Then, it follows that after the second gradient step,

$$\left\| W_O - 2\eta\bar{B} \right\|_F \le \frac{\eta^2}{\sqrt{|V|}} \tag{44}$$

Now, we consider the gradient with respect to $V^{(1)}$. By Lemma D.1, we have that

$$\frac{\partial \mathcal{L}}{\partial V^{(1)}} = \frac{-1}{NT} \sum_{i=1}^{N} \eta X_i^\top A_i^{(1)\top} (Y_i - \mathcal{S}(F_\theta(X_i)))\bar{B}^\top \tag{45}$$

and since $W^{(1)}, P^{(1)} = \mathbf{0}$, $A_i^{(1)} = A_0$ where the $t$th row of $A_0$ has the first $t$ elements equal to $1/t$ and the rest equal to 0. Then, by equation 41, we have that

$$\left\| \eta X_i^\top A_0^\top (Y_i - \mathcal{S}(F_\theta(X_i)))\bar{B}^\top - \eta X_i^\top A_0^\top (Y_i - U_O)\bar{B}^\top \right\|_F \le \frac{\eta^2 T}{\sqrt{|V|}} \|A_0\| \tag{46}$$

Then, using the discrete Hardy's inequality with $p = 2$, we have that $\|A_0\| \le 2$ and

$$\left\| \frac{\partial \mathcal{L}}{\partial V^{(1)}} + \frac{1}{NT} \sum_{i=1}^{N} \eta X_i^\top A_0^\top (Y_i - U_O)\bar{B}^\top \right\|_F \le \frac{2\eta^2}{\sqrt{|V|}} \tag{47}$$

Now, we will analyze

$$\frac{1}{NT} \sum_{i=1}^{N} \eta X_i^\top A_0^\top (Y_i - U_O)\bar{B}^\top \tag{48}$$

Since $\eta\bar{B}$ is independent of $i$, we can move it outside the sum and we can analyze

$$\frac{1}{NT} \sum_{i=1}^{N} X_i^\top A_0^\top (Y_i - U_O) \tag{49}$$

We start by considering the form of $X_i^\top A_0^\top$. Since the $t$th row of $A_0$ has $1/t$ as the first $t$ elements and zeros for all other elements, we have that the $j$th element of the $t$th column of $X_i^\top A_0^\top$ is

$$\frac{\gamma_i(e_j, t)}{t} \tag{50}$$

where $e_j$ represents the $j$th token in the vocabulary and $\gamma_i(e_j, t)$ is the number of occurrences of $e_j$ in the first $t$ tokens of $X_i$. Letting $\Phi' = \frac{1}{NT} \sum_{i=1}^{N} X_i^\top A_0^\top Y_i$, we have that

$$\Phi'_{jk} = \frac{1}{NT} \sum_{i=1}^{N} \sum_{t=1}^{T} \mathbb{1}(X_i^{[t+1]} = e_k) \frac{\gamma_i(e_j, t)}{t} \tag{51}$$

Swapping the order of the sums, we have

$$\Phi'_{jk} = \frac{1}{NT} \sum_{t=1}^{T} \frac{1}{t} \sum_{i=1}^{N} \mathbb{1}(X_i^{[t+1]} = e_k)\gamma_i(e_j, t) \tag{52}$$

Then, as $\gamma_i(e_j, t) = \sum_{m=1}^{t} \mathbb{1}(X_i^{[m]} = e_j)$, we have that

$$\Phi'_{jk} = \frac{1}{T} \sum_{t=1}^{T} \frac{1}{t} \sum_{m=1}^{t} \frac{1}{N} \sum_{i=1}^{N} \mathbb{1}(X_i^{[t+1]} = e_k, X_i^{[m]} = e_j) \tag{53}$$

Then, as $\frac{1}{N}\sum_{i=1}^{N}$ corresponds to an average over the dataset, the average over $N$ corresponds to the empirical probability of having a sequence with the $t+1$th token equal to $e_k$ and the $m$th token equal to $e_j$, which we will denote as $\mathcal{P}(x_{t+1}=e_k, x_m=e_j)$. Then, this gives

$$\Phi'_{jk} = \frac{1}{T}\sum_{t=1}^{T}\frac{1}{t}\sum_{m=1}^{t}\mathcal{P}(x_{t+1}=e_k, x_m=e_j) \tag{54}$$

Then, we have an average over $m \in [t]$ which results in the average probability that the $t+1$th token is $e_k$ and $e_j$ is in the first $t$ tokens. We will denote this as $\mathcal{P}(e_j \in x_{1:t}, x_{t+1}=e_k)$. This gives

$$\Phi'_{jk} = \frac{1}{T}\sum_{t=1}^{T}\mathcal{P}(e_j \in x_{1:t}, x_{t+1}=e_k) \tag{55}$$

This probability of $e_j$ being in the prefix of $x_{t+1}=e_k$ is averaged over the different positions of $e_k$ to get an average probability that $e_j$ is in the prefix given that $e_k$ is the current token, which we will denote as $\mathcal{P}(e_j \to e_k)$ and

$$\Phi'_{jk} = \mathcal{P}(e_j \to e_k) \tag{56}$$

Now, we consider $U_P = \frac{1}{NT}\sum_{i=1}^{N}X_i^\top A_0^\top U_O$. Then, we have that

$$U_{P_{jk}} = \frac{1}{NT}\sum_{i=1}^{N}\sum_{t=1}^{T}\frac{\gamma_i(e_j, t)}{|V|t} \tag{57}$$

Rearranging the sum and decomposing $\gamma_i$, we get

$$U_{P_{jk}} = \frac{1}{T}\sum_{t=1}^{T}\frac{1}{t|V|}\sum_{m=1}^{t}\mathcal{P}(x_m=e_j) \tag{58}$$

Then, if we consider the average over positions $m$ and $t$, we get the average probability that $e_j$ is in the first $t$ tokens over all $t$ multiplied by $1/|V|$. We can notice that the sum of the $j$th row of $U_P$ and $\Phi'$ are the same. Then, setting

$$\bar{\Phi}' = \Phi' - U_P \tag{59}$$

we have

$$\left\|\frac{\partial \mathcal{L}}{\partial V^{(1)}} + \eta\bar{\Phi}'\bar{B}^\top\right\|_F \leq \frac{2\eta^2}{\sqrt{|V|}} \tag{60}$$

Then, it follows that after two gradient steps,

$$\left\|V^{(1)} - \eta^2\bar{\Phi}'\bar{B}^\top\right\|_F \leq \frac{2\eta^3}{\sqrt{|V|}} \tag{61}$$

Defining $\bar{\Phi} = \bar{\Phi}'^\top$, we have

$$\left\|V^{(1)} - \eta^2\bar{\Phi}^\top\bar{B}^\top\right\|_F \leq \frac{2\eta^3}{\sqrt{|V|}} \tag{62}$$

This completes the proof. □

**Lemma D.6** (Third Gradient Step). *Under the setting described, and with $\eta \leq 1/8$, $|V| \geq 500$, after three gradient steps with $\eta$, we have that*

$$\left\|W_O - 3\eta\bar{B}\right\|_F \leq 3\eta^2 \tag{63}$$

$$\left\|V^{(1)} - 3\eta^2\bar{\Phi}^\top\bar{B}^\top\right\|_F \leq 2\eta^3 \tag{64}$$

$$\left\|W^{(1)} - 2\eta^4\bar{Q}\right\|_F \leq 2\eta^5 T \tag{65}$$

$$\left\|P^{(1)} - 2\eta^4\Delta\right\|_F \leq 2\eta^5 T \tag{66}$$

*Proof.* First, we start with bounding the norm of $W_O, V^{(1)}, A_0$. We have that

$$\|W_O\|_F \le 2\eta + \frac{\eta^2}{\sqrt{|V|}} \tag{67}$$

$$\left\|V^{(1)}\right\| \le \left(2\eta^2 + \frac{2\eta^3}{\sqrt{|V|}}\right) \tag{68}$$

$$\|A_0\| \le 2 \tag{69}$$

Now, we consider the deviation of the output from the uniform distribution. We start by bounding the norm of $X_i + A_0 X_i V^{(1)}$

$$\left\|X_i + A_0 X_i V^{(1)}\right\| \le \left(1 + 5\eta^2\right)\sqrt{T} \tag{70}$$

Then, we can upper bound the norm of $F_\theta(X_i)$ as

$$\|F_\theta(X_i)\|_F \le \frac{5\eta}{2}\left(1 + 5\eta^2\right)\sqrt{T} \le 4\eta\sqrt{T}, \tag{71}$$

where the last inequality uses $\eta \le 1/8$. Let

$$H_i = X_i + A_0 X_i V^{(1)}.$$

For each row $t$, write $f_{i,t} = F_\theta(X_i)^{[t,:]} = H_i^{[t,:]} W_O$. Using the bounds above,

$$\left\|H_i^{[t,:]}\right\| \le 1 + 5\eta^2, \qquad \|W_O\| \le \|W_O\|_F \le \frac{5\eta}{2}.$$

Therefore

$$\|f_{i,t}\|_2 \le \frac{5\eta}{2}(1 + 5\eta^2).$$

Hence, since $\eta \le 1/8$,

$$\mathrm{range}(f_{i,t}) \le 2\|f_{i,t}\|_2 \le 1.$$

By Lemma D.2, applied rowwise,

$$\left\|\mathcal{S}(f_{i,t}) - \frac{1}{|V|}\mathbf{1}\right\| \le \frac{e}{|V|}\|f_{i,t}\|.$$

Summing over rows gives

$$\|\mathcal{S}(F_\theta(X_i)) - U_O\|_F \le \frac{e}{|V|}\|F_\theta(X_i)\|_F \le \frac{4e\eta\sqrt{T}}{|V|} \le \frac{4\eta\sqrt{T}}{\sqrt{|V|}}, \tag{72}$$

where the last inequality uses $|V| \ge 8$. Then, it follows that

$$\left\|\frac{\partial \mathcal{L}}{\partial W_O} + \bar{B}\right\|_F \le \frac{1}{NT}\sum_{i=1}^{N}\left(\frac{8\eta T}{\sqrt{|V|}} + 5\eta^2 T\right) \le 2\eta \tag{73}$$

and

$$\left\|W_O - 3\eta\bar{B}\right\|_F \le \frac{\eta^2}{\sqrt{|V|}} + 2\eta^2 \le 3\eta^2 \tag{74}$$

Now, we consider the gradient with respect to $V^{(1)}$. Since $\|Y_i - \mathcal{S}(F_\theta(X_i))\| \le \sqrt{2T}$, we have that

$$\left\|\frac{\partial \mathcal{L}}{\partial V^{(1)}} + 2\eta\bar{\Phi}^\top\bar{B}^\top\right\|_F \le 2\left(\frac{4\eta}{\sqrt{|V|}}\frac{5\eta}{2} + \frac{\eta^2}{\sqrt{|V|}}\right) \le \frac{22\eta^2}{\sqrt{|V|}} \le \eta^2 \tag{75}$$

Then, we have that after the third step,

$$\left\|V^{(1)} - 3\eta^2\bar{\Phi}^\top\bar{B}^\top\right\|_F \le 2\eta^3 \tag{76}$$

Now, we consider the gradient with respect to $W^{(1)}, P^{(1)}$ which according to Lemma 1.1 are

$$\frac{\partial \mathcal{L}}{\partial W^{(1)}} = \frac{-1}{NT} \sum_{i=1}^{N} X_i^\top ein_{tjk,tk \to tj}(J_i, R_i W_O^\top V^{(1)\top} X_i^\top) X_i \tag{77}$$

$$\frac{\partial \mathcal{L}}{\partial P^{(1)}} = \frac{-1}{NT} ein_{tjk,jk \to t}\left(D, \sum_{i=1}^{N} ein_{tjk,tk \to tj}(J_i, R_i W_O^\top V^{(1)\top} X_i^\top)\right) \tag{78}$$

We start by analyzing $R_i W_O^\top V^{(1)\top} X_i^\top$. First, We will use that $\|Y_i - U_O\| \le \sqrt{T}$ and

$$\|\bar{\Phi}\| \le \frac{1}{T} \max_i \|X_i\| \|A_0\| \|Y_i - U_O\| \le \frac{2}{T} T = 2 \tag{79}$$

We know by the previous lemma and the bound on the deviation of the output that

$$\left\| R_i W_O^\top V^{(1)\top} X_i^\top - 2\eta^3 (Y_i - U_O) \bar{B}^\top \bar{B} \bar{\Phi} X_i^\top \right\|_F \le \frac{25\eta^4 T}{\sqrt{|V|}} + \frac{5T\eta^4}{2\sqrt{|V|}} + \frac{4\eta^4 T}{\sqrt{|V|}} \le \frac{3\eta^4 T}{2} \tag{80}$$

We start by considering the structure of $Y_i \bar{B}^\top \bar{B} \bar{\Phi} X_i^\top$. We first consider the simpler multiplication of $e_j \bar{B}^\top \bar{B} \bar{\Phi} e_k^\top$. First, we define $\Sigma_{\bar{B}} = \bar{B}^\top \bar{B}$ which has $\Sigma_{\bar{B}mn} = \sum_{l=1}^{|V|} \alpha_l^2 \bar{P}_{lm} \bar{P}_{ln}$ where $\alpha_l$ is the relative frequency of token $l$ and $\bar{P}_l$ is the average next-token distribution of token $e_l$ centered at 0. This corresponds to a similarity measure of the previous tokens of $e_m$ and $e_n$ with common tokens more heavily weighted. Then, we have that

$$e_j \Sigma_{\bar{B}} \bar{\Phi} e_k^\top = \sum_{m=1}^{|V|} \Sigma_{\bar{B}jm} \bar{\mathcal{P}}(e_m \to e_k) \tag{81}$$

where $\bar{\mathcal{P}}(e_m \to e_k)$ is the probability that $e_m$ is in the prefix of $e_k$ centered at 0. We can then interpret the each element $(\Sigma_{\bar{B}} \bar{\Phi})_{jk}$ as a measure of assocation between token $j$ and $k$ based on a two-step chain of (interchangeability mapping, suffix token mapping). Essentially, how often does token $e_k$ succeed token $e_j$ and similar tokens. We will let $\bar{G} = \Sigma_{\bar{B}} \bar{\Phi}$. Now, we can consider $2\eta^3 Y_i \bar{G} X_i^\top$. This results in a $T \times T$ matrix where the $jk$-th element is $\tilde{\mathcal{P}}(X_i^{[k]} \to_2 X_i^{[j+1]})$ and we will denote this as $g_{ijk}$. Then, $2\eta^3 (Y_i - U_O) \bar{G} X_i^\top$ will have elements centered to have row sums of 0 and we will let the centered elements be $\bar{g}_{ijk}$. Then, we consider the einsum of $J$ and $2\eta^3 (Y_i - U_O) \bar{G} X_i^\top$. The $t$th column of the resulting matrix will the be product of $J_t$ and the $t$th column of $2\eta^3 (Y_i - U_O) \bar{G} X_i^\top$. This results in the $t$th row having the form

$$2\eta^3 \left( \begin{bmatrix} \frac{\bar{g}_{i1t}}{t} \\ \vdots \\ \frac{\bar{g}_{itt}}{t} \\ 0 \\ \vdots \\ 0 \end{bmatrix} - \begin{bmatrix} \frac{\mu_{g,it}}{t} \\ \vdots \\ \frac{\mu_{g,it}}{t} \\ 0 \\ \vdots \\ 0 \end{bmatrix} \right)^\top \tag{82}$$

The $t$th row is $2\eta^3 \bar{g}_{ijt}$ for $1 \le j \le t$ weighted by $1/t$ and centered to have a row sum of 0 and we will refer to the centered and weighted elements $2\eta^3 q_{ijt}$ and the resulting matrix $Q_i$. Letting $\bar{Q} = \frac{1}{NT} \sum_{i=1}^{N} X_i^\top Q_i X_i$, we have that

$$\left\| \frac{\partial \mathcal{L}}{\partial W^{(1)}} + 2\eta^3 \bar{Q} \right\|_F \le 2\eta^4 T \tag{83}$$

where we have used that the squared Frobenius norm of the einsum is the sum of the norms of each column and that $\|J_t\| = 1/t$. Then, it follows that

$$\left\| W^{(1)} - 2\eta^4 \bar{Q} \right\|_F \le 2\eta^5 T \tag{84}$$

Now, we consider the gradient with respect to each element of $P^{(1)}$

$$\left\| \frac{\partial \mathcal{L}}{\partial P_m^{(1)}} + \frac{2\eta^3}{NT} \sum_{i=1}^{N} \text{Tr}(D_{-m} Q_i) \right\|_F \le 2\eta^4 \sqrt{T} \tag{85}$$

Since the trace is a linear function, we let $\Delta_m = \mathrm{Tr}(D_{-m}\frac{1}{NT}\sum_{i=1}^{N}Q_i)$ and let $\Delta$ be the vector consisting of $\Delta_m$, and we have

$$\left\|\frac{\partial\mathcal{L}}{\partial P^{(1)}} + 2\eta^3\Delta\right\|_F \leq 2\eta^4 T \tag{86}$$

and it follows that

$$\left\|P^{(1)} - 2\eta^4\Delta\right\|_F \leq 2\eta^5 T \tag{87}$$

This completes the proof. □

**Theorem D.7** (Early Stage Features). *Under the setting described, for $s \leq \eta^{-1}\frac{3}{8T^{3/8}}$, for $T \geq 3, |V| \geq 500$, we have that after $s$ gradient descent steps with learning rate $\eta$,*

$$\left\|W_O - s\eta\bar{B}\right\|_F \leq 3s^2\eta^2 \tag{88}$$

$$\left\|V^{(1)} - \binom{s}{2}\eta^2\bar{\Phi}^\top\bar{B}^\top\right\|_F \leq 4s^3\eta^3 \tag{89}$$

$$\left\|W^{(1)} - \left(3\binom{s}{4} + 2\binom{s}{3}\right)\eta^4\bar{Q}\right\|_F \leq 6s^5\eta^5 T \tag{90}$$

$$\left\|P^{(1)} - \left(3\binom{s}{4} + 2\binom{s}{3}\right)\eta^4\Delta\right\|_F \leq 6s^5\eta^5 T \tag{91}$$

*Proof.* We will prove the result by induction. The previous lemmas form the base case. The first phase will be bounding the deviation of the output from the uniform distribution after $s$ gradient steps. To start, we bound the norm of $A_i$, the resulting attention mapping for $X_i$. For the $t$th row, define

$$z_{i,t} := (X_i W^{(1)} X_i^\top + \mathrm{DM}(P^{(1)}))[t,:t].$$

By the inductive hypotheses for $W^{(1)}$ and $P^{(1)}$,

$$\|z_{i,t}\|_\infty \leq \left(6\binom{s}{4} + 4\binom{s}{3}\right)\eta^4 + 12s^5\eta^5 T.$$

Using $6\binom{s}{4} + 4\binom{s}{3} \leq 2s^4$ and $s\eta \leq 3/(8T^{3/8})$, the right-hand side is bounded by an absolute constant smaller than $1/2$. Hence

$$\mathrm{range}(z_{i,t}) \leq 2\|z_{i,t}\|_\infty \leq 1.$$

By Lemma D.2,

$$\|(A_i - A_0)[t,:]\| \leq \frac{e}{t}\|z_{i,t}\|.$$

Moreover,

$$\|z_{i,t}\| \leq \sqrt{t}\left(\left(6\binom{s}{4} + 4\binom{s}{3}\right)\eta^4 + 12s^5\eta^5 T\right).$$

Therefore

$$\|(A_i - A_0)[t,:]\| \leq e\left(6\binom{s}{4} + 4\binom{s}{3}\right)\eta^4 + \frac{12e\, s^5\eta^5 T}{\sqrt{t}}.$$

Then, summing the upper bounds on the squared norms of each row, and using that $\sum_{q=1}^{r}1/q \leq 1 + \log r$ we have

$$\|A_i - A_0\|_F \leq e\left(6\binom{s}{4} + 4\binom{s}{3}\right)\eta^4\sqrt{T} + 12es^5\eta^5 T\sqrt{1 + \log T} \tag{92}$$

Then, we have that

$$\left\|A_i^{(1)}\right\| \leq 2 + e\left(6\binom{s}{4} + 4\binom{s}{3}\right)\eta^4\sqrt{T} + 12es^5\eta^5 T\sqrt{1 + \log T} \tag{93}$$

Then, upper bounding $6\binom{s}{4} + 4\binom{s}{3}$ by $2s^4$, we have

$$\left\|A_i^{(1)}\right\| \leq 2 + 2es^4\eta^4\sqrt{T} + 12es^5\eta^5 T\sqrt{1 + \log T} \tag{94}$$

Now, using that $s\eta \le \frac{3}{8T^{3/8}}$, we have that

$$2es^4\eta^4\sqrt{T} + 12es^5\eta^5T\sqrt{1+\log T} \le 2s\eta \tag{95}$$

and

$$\left\|A_i^{(1)}\right\| \le 2 + 2s\eta \tag{96}$$

Now, we bound the norm of $V^{(1)}$ which by the inductive hypothesis we have is at most

$$\binom{s}{2}\eta^2 2\sqrt{2} + 4s^3\eta^3 \tag{97}$$

which is at most

$$s^2\eta^2\sqrt{2} + 4s^3\eta^3 \tag{98}$$

and since $s\eta \le \frac{1}{3}$,

$$\left\|V^{(1)}\right\|_F \le 4s^2\eta^2 \tag{99}$$

Then, we have that

$$\left\|X_i + A_i^{(1)}X_iV^{(1)}\right\|_F \le \sqrt{T}\left(1 + 16s^2\eta^2\right) \tag{100}$$

Since $s\eta \le \frac{3}{8T^{3/8}}$ and by the inductive hypothesis we have that

$$\|F_\theta(X_i)\|_F \le 2\sqrt{T}\left(s\eta + 3s^2\eta^2\right) \tag{101}$$

Since $s\eta \le \frac{3}{8T^{3/8}}$, we have

$$\|F_\theta(X_i)\|_F \le 4s\eta\sqrt{T} \tag{102}$$

Let

$$H_i := X_i + A_i^{(1)}X_iV^{(1)}.$$

For each row $t$, write

$$f_{i,t} := F_\theta(X_i)^{[t,:]} = H_i^{[t,:]}W_O.$$

Since $A_i^{(1)}[t,:]X_i$ is a convex combination of one-hot vectors,

$$\left\|H_i^{[t,:]}\right\| \le 1 + \left\|V^{(1)}\right\| \le 1 + 4s^2\eta^2.$$

Also, by the inductive hypothesis and $\left\|\bar{B}\right\|_F \le 1$,

$$\|W_O\| \le \|W_O\|_F \le s\eta + 3s^2\eta^2.$$

Thus

$$\|f_{i,t}\| \le (1 + 4s^2\eta^2)(s\eta + 3s^2\eta^2).$$

Using $s\eta \le 3/(8T^{3/8})$ and $T \ge 3$, this is bounded by 1, and hence

$$\mathrm{range}(f_{i,t}) \le 2\|f_{i,t}\| \le 2.$$

By Lemma D.2, applied row-wise,

$$\left\|\mathcal{S}(f_{i,t}) - \frac{1}{|V|}\mathbf{1}\right\| \le \frac{e^2}{|V|}\|f_{i,t}\|.$$

Summing over rows gives

$$\|\mathcal{S}(F_\theta(X_i)) - U_O\|_F \le \frac{e^2}{|V|}\|F_\theta(X_i)\|_F \le \frac{4e^2s\eta\sqrt{T}}{|V|}.$$

Since $|V| \ge 500$, we have $e^2/\sqrt{|V|} \le 1$, and therefore

$$\|\mathcal{S}(F_\theta(X_i)) - U_O\|_F \le 4s\eta\sqrt{\frac{T}{|V|}}.$$

Now, we will utilize the bound on the deviation of the output from the uniform distribution as well to perform the inductive step for $W_O$. We have based on the bound that after the $(s+1)$th step

$$\left\| \frac{\partial \mathcal{L}}{\partial W_O} + \bar{B} \right\|_F \leq \frac{8s\eta}{\sqrt{|V|}} + 12s^2\eta^2 \leq 6s\eta \tag{103}$$

Then, after $(s+1)$ steps, we have that

$$\left\| W_O - (s+1)\eta\bar{B} \right\|_F \leq 3\eta^2 s^2 + 6s\eta^2 \leq 3\eta^2(s+1)^2 \tag{104}$$

Now, we perform the inductive step for $V^{(1)}$ using the bound on the deviation of the attention pattern and on the output deviation. We have that after the $(s+1)$th step

$$\left\| \frac{\partial \mathcal{L}}{\partial V^{(1)}} + s\eta\bar{\Phi}^\top\bar{B}^\top \right\|_F \leq \| R_i - (Y_i - U_O) \|_F \left\| A_i^{(1)} \right\| \| X_i \| \| W_O \|$$
$$+ \| (Y_i - U_O) \| \left\| A_i^{(1)} - A_0 \right\|_F \| X_i \| \| W_O \|$$
$$+ \| (Y_i - U_O) \| \| A_0 \| \| X_i \| \left\| W_O - s\eta\bar{B} \right\|_F \tag{105}$$

Applying upper bounds and using that $|V| \geq 500$, we have

$$\left\| \frac{\partial \mathcal{L}}{\partial V^{(1)}} + s\eta\bar{\Phi}^\top\bar{B}^\top \right\|_F \leq \frac{1}{T}\left( \frac{24s\eta}{\sqrt{|V|}}T\eta s + 4Ts^2\eta^2 + 6Ts^2\eta^2 \right) \leq 12s^2\eta^2 \tag{106}$$

Then, after $(s+1)$ steps, we have that

$$\left\| V^{(1)} - \binom{s+1}{2}\eta^2\bar{\Phi}^\top\bar{B}^\top \right\|_F \leq 4s^3\eta^3 + 12s^2\eta^3 \leq 4(s+1)^3\eta^3 \tag{107}$$

Now, we perform the inductive step for $W^{(1)}$ utilizing the earlier bounds on the output and attention pattern deviations. We start by bounding the deviation between $\frac{s^3-s^2}{2}\eta^3\Sigma_{\bar{B}}\bar{\Phi}$ and $W_O^\top V^{(1)\top}$. By the inductive hypothesis and $2 \leq \sqrt{|V|}$, we have that

$$\left\| W_O^\top V^{(1)\top} - \frac{s^3-s^2}{2}\eta^3\Sigma_{\bar{B}}\bar{\Phi} \right\|_F \leq 8s^4\eta^4 + 3\sqrt{2}s^4\eta^4 \leq 13s^4\eta^4 \tag{108}$$

Then, for each $R_i W_O^\top V^{(1)\top} X_i^\top$ since $|V| \geq 500$, we have that

$$\left\| R_i W_O^\top V^{(1)\top} X_i^\top - \frac{s^3-s^2}{2}\eta^3(Y_i - U_O)\Sigma_{\bar{B}}\bar{\Phi}X_i^\top \right\|_F \leq \frac{20s^4\eta^4 T}{\sqrt{|V|}} + 13s^4\eta^4 T \leq 14s^4\eta^4 T \tag{109}$$

Now, in order to consider the deviation of the einsum of $J$ and $\frac{s^3-s^2}{2}\eta^3(Y_i - U_O)\Sigma_{\bar{B}}\bar{\Phi}X_i^\top$, we need to first bound the deviation of the Jacobian of the current attention pattern from $J$. We do so by considering the deviation for each $J_t$. As proven earlier, we have that

$$\| (A_i - A_0)[t,:] \| \leq e\left( 6\binom{s}{4} + 4\binom{s}{3} \right)\eta^4 + \frac{12es^5\eta^5 T}{\sqrt{t}} \tag{110}$$

Then, we have that for the current Jacobian for the sample $X_i$ corresponding to the $t$th row which will call $J_{t,i}$

$$J_{t,i} - J_t = \text{Diag}(A_i[t,:] - A_0[t,:]) - A_0[t,:](A_i[t,:] - A_0[t,:])^\top - (A_i[t,:] - A_0[t,:])A_0[t,:]^\top$$
$$- (A_i[t,:] - A_0[t,:])(A_i[t,:] - A_0[t,:])^\top \tag{111}$$

and it follows then that for $t \geq 2$

$$\| J_{t,i} - J_t \|_2 \leq \| A_i[t,:] - A_0[t,:] \|_\infty + \frac{2}{\sqrt{t}}\| A_i[t,:] - A_0[t,:] \|_2 + \| A_i[t,:] - A_0[t,:] \|_2^2 \tag{112}$$

Then, as

$$\| A_i[t,:] - A_0[t,:] \|_\infty \leq \| A_i[t,:] - A_0[t,:] \|_2 \tag{113}$$

we have that

$$\|J_{t,i} - J_t\|_2 \leq \left(e\left(6\binom{s}{4} + 4\binom{s}{3}\right)\eta^4 + \frac{12es^5\eta^5 T}{\sqrt{t}}\right)$$
$$\left(1 + \frac{2}{\sqrt{t}} + \left(6\binom{s}{4} + 4\binom{s}{3}\right)\eta^4 + \frac{12s^5\eta^5 T}{\sqrt{t}}\right) \quad (114)$$

Since $J_{1,i}$ is always all zeros, we can ignore this term and for $t \geq 2$, we have that as $s\eta \leq \frac{3}{8T^{3/8}}$,

$$\|J_{t,i} - J_t\|_2 \leq 5s^2\eta^2 \quad (115)$$

Then, we have that

$$\left\|\frac{s^3 - s^2}{2}\eta^3 Q_i - \text{ein}_{tjk,tk\to tj}(J_i, R_i W_O^\top V^{(1)\top} X_i^\top)\right\|_F$$
$$\leq \|J_i - J\|_2 \left\|\frac{s^3 - s^2}{2}\eta^3 (Y_i - U_O)\Sigma_{\bar{B}}\bar{\Phi} X_i^\top\right\|_F \quad (116)$$
$$+ \|J_i\|_2 \left\|R_i W_O^\top V^{(1)\top} X_i^\top - \frac{s^3 - s^2}{2}\eta^3 (Y_i - U_O)\Sigma_{\bar{B}}\bar{\Phi} X_i^\top\right\|_F$$

Then, as $\|J_t\|_2 = \frac{1}{t}$, $\|J_i\|_2 \leq \frac{3}{2} + 5s^2\eta^2\sqrt{T} \leq 2$, and $s\eta \leq \frac{3}{8T^{3/8}}$, we have that

$$\left\|\frac{s^3 - s^2}{2}\eta^3 Q_i - \text{ein}_{tjk,tk\to tj}(J_i, R_i W_O^\top V^{(1)\top} X_i^\top)\right\|_F \leq 5\sqrt{2}s^5\eta^5 T + 28s^4\eta^4 T \leq 30s^4\eta^4 T$$
$$(117)$$

Then, we have that

$$\left\|\frac{\partial\mathcal{L}}{\partial W^{(1)}} + \frac{s^3 - s^2}{2}\eta^3\bar{Q}\right\|_F \leq 30s^4\eta^4 T \quad (118)$$

Then, we have that after $(s + 1)$ steps,

$$\left\|W^{(1)} - \left(3\binom{s+1}{4} + 2\binom{s+1}{3}\right)\eta^4\bar{Q}\right\|_F \leq 6s^5\eta^5 T + 30s^4\eta^5 T \leq 6(s+1)^5\eta^5 T \quad (119)$$

Finally, as we have the bound on the deviation from $Q_i$, we have that for $P^{(1)}$,

$$\left\|\frac{\partial\mathcal{L}}{\partial P^{(1)}} + \frac{s^3 - s^2}{2}\eta^3\Delta\right\|_F \leq 30s^4\eta^4 T \quad (120)$$

and that after $(s + 1)$ steps,

$$\left\|P^{(1)} - \left(3\binom{s+1}{4} + 2\binom{s+1}{3}\right)\eta^4\Delta\right\|_F \leq 6s^5\eta^5 T + 30s^4\eta^5 T \leq 6(s+1)^5\eta^5 T \quad (121)$$

$\square$

### D.2 Proof of Multi-layer Theorem

**Lemma D.8** (General Gradient Form). *Under the setting described, defining*

$$S_i^{(l)} = \text{ein}_{tjk,\, tk\to tj}\left(J_i^{(l)}, G_i^{(l)} V^{(l)\top} h_i^{(l-1)\top}\right), \quad (122)$$

$$G_i^{(l-1)} = G_i^{(l)} + A_i^{(l)\top} G_i^{(l)} V^{(l)\top} + S_i^{(l)} h_i^{(l-1)} W^{(l)\top} + S_i^{(l)\top} h_i^{(l-1)} W^{(l)}, \quad (123)$$

*with*

$$G_i^{(L)} = R_i W_O^\top \quad (124)$$

*we have that*

$$\frac{\partial\mathcal{L}}{\partial W_O} = \frac{-1}{NT}\sum_{i=1}^N h_i^{(L)\top} R_i, \quad (125)$$

$$\frac{\partial \mathcal{L}}{\partial V^{(l)}} = \frac{-1}{NT} \sum_{i=1}^{N} h_i^{(l-1)\top} A_i^{(l)\top} G_i^{(l)}, \tag{126}$$

$$\frac{\partial \mathcal{L}}{\partial W^{(l)}} = \frac{-1}{NT} \sum_{i=1}^{N} h_i^{(l-1)\top} S_i^{(l)} h_i^{(l-1)}, \tag{127}$$

$$\frac{\partial \mathcal{L}}{\partial P^{(l)}} = \frac{-1}{NT} ein_{tjk,\, jk \to t}\left(D, \sum_{i=1}^{N} S_i^{(l)}\right), \tag{128}$$

where $A_i^{(l)} = \mathcal{S}(\text{Mask}(h_i^{(l-1)} W^{(l)} h_i^{(l-1)\top} + \text{D}_{\text{m}}(P^{(l)})))$, $R_i = Y_i - \mathcal{S}(F_\theta(X_i))$, $J_i^{(l)} \in \mathbb{R}^{T \times T \times T}$ with $J_{i,t}^{(l)} = \text{Diag}(A_i^{(l)[t]}) - A_i^{(l)[t]\top} A_i^{(l)[t]}$ being the Jacobian of the softmax function at the lth attention layer for the tth token in the sequence, $D \in \mathbb{R}^{T \times T \times T}$ with $D_t$ being a matrix with ones along the $(-t+1)$th sub-diagonal and zeros elsewhere, and ein denotes an Einstein summation.

*Proof.* Throughout the proof, we factor out the global multiplier $-1/(NT)$ and define $G_i^{(l)}$ to be the resulting backpropagated residual at layer $l$. Thus the actual gradient with respect to $h_i^{(l)}$ is

$$\frac{\partial \mathcal{L}}{\partial h_i^{(l)}} = -\frac{1}{NT} G_i^{(l)}.$$

We begin by considering the derivative of the loss with respect to $F_\theta(X_i)^{[t]}$, which is

$$\frac{\partial \mathcal{L}}{\partial F_\theta(X_i)^{[t]}} = \mathcal{S}(F_\theta(X_i)^{[t]}) - Y_i^{[t]} = -R_i^{[t]} \tag{129}$$

Since

$$\frac{\partial F_\theta(X_i)^{[t]}}{\partial W_O} = h_i^{(L)[t]} \tag{130}$$

it follows that

$$\frac{\partial \mathcal{L}}{\partial W_O} = \frac{-1}{NT} \sum_{i=1}^{N} h_i^{(L)\top} R_i \tag{131}$$

We now consider the gradient through each attention layer in terms of the current and previous layer embeddings $h_i^{(l-1)}, h_i^{(l)}$. Let $h = h_i^{(l-1)}$, $A = A_i^{(l)}$, $G = G_i^{(l)}$, and $V = V^{(l)}$. The residual contribution to the gradient with respect to $V^{(l)}$ is

$$(Ah)^\top G = h^\top A^\top G.$$

Therefore, after restoring the global factor and summing over $i$,

$$\frac{\partial \mathcal{L}}{\partial V^{(l)}} = -\frac{1}{NT} \sum_{i=1}^{N} h_i^{(l-1)\top} A_i^{(l)\top} G_i^{(l)}.$$

The gradient for $M = Ah$ is $\delta_M = GV^{(l)\top}, \delta_A = \delta_M h^\top, \delta_h^{(1)} = A^\top \delta_M$. Next, through the row-wise softmax, each row Jacobian is

$$J_{i,t}^{(l)} = \text{Diag}(A_i^{(l)[t]}) - A_i^{(l)[t]\top} A_i^{(l)[t]} \tag{132}$$

Stacking these gives a tensor $J_i^{(l)}$. Applying it row-wise to $\delta_A$ gives

$$S_i^{(l)} = ein_{tjk,\, tk \to tj}\left(J_i^{(l)}, G_i^{(l)} V^{(l)\top} h_i^{(l-1)\top}\right) \tag{133}$$

Finally, back-propagating through $\tilde{A} = hW^{(l)} h^\top + \text{DM}(P^{(l)})$, the residual contributions are

$$h^\top S_i^{(l)} h$$

for $W^{(l)}$ and

$$ein_{tjk,\, jk \to t}(D, S_i^{(l)})$$

for $P^{(l)}$. Summing over $i$ and normalizing,

$$\frac{\partial \mathcal{L}}{\partial W^{(l)}} = \frac{-1}{NT} \sum_{i=1}^{N} h_i^{(l-1)\top} S_i^{(l)} h_i^{(l-1)} \tag{134}$$

$$\frac{\partial \mathcal{L}}{\partial P^{(l)}} = \frac{-1}{NT} \operatorname{ein}_{tjk,\, jk \to t} \left( D, \sum_{i=1}^{N} S_i^{(l)} \right) \tag{135}$$

Collecting all contributions to $h_i^{(l-1)}$ gives

$$G_i^{(l-1)} = G_i^{(l)} + A_i^{(l)\top} G_i^{(l)} V^{(l)\top} + S_i^{(l)} h_i^{(l-1)} W^{(l)\top} + S_i^{(l)\top} h_i^{(l-1)} W^{(l)} \tag{136}$$

Since

$$\frac{\partial \mathcal{L}}{\partial F_\theta(X_i)} = \mathcal{S}(F_\theta(X_i)) - Y_i = -R_i,$$

after factoring out $-1/(NT)$, the residual at the last layer is

$$G_i^{(L)} = R_i W_O^\top.$$

We can inductively apply the recurrence and collecting the per-layer parameter derivatives gives the desired expressions for $\partial \mathcal{L}/\partial W_O$, $\partial \mathcal{L}/\partial V^{(l)}$, $\partial \mathcal{L}/\partial W^{(l)}$, and $\partial \mathcal{L}/\partial P^{(l)}$. $\qquad\square$

**Lemma D.9** (First Step, Multi-Layer Zero-Initialization). *Under the setting described, after one gradient step, we have that*

$$W_O = \eta(\bar{B}) \tag{137}$$
$$W^{(l)}, V^{(l)}, P^{(l)} = \mathbf{0} \tag{138}$$

*for $1 \leq l \leq L$ where $\bar{B}$ is a $|V| \times |V|$ matrix where the $j$th row is the average next-token distribution of the $j$th token in the vocabulary weighted by the relative frequency of token $j$ across the dataset and centered to have the row sum be 0.*

*Proof.* By Lemma D.8,

$$\frac{\partial \mathcal{L}}{\partial W_O} = \frac{-1}{NT} \sum_{i=1}^{N} h_i^{(L)\top} R_i = \frac{-1}{NT} \sum_{i=1}^{N} X_i^\top (Y_i - U_O) = -(B - U) \equiv -\bar{B} \tag{139}$$

where $B$ and $U$ are defined the same as in the one-layer case. A single gradient step gives

$$W_O = -\eta \frac{\partial \mathcal{L}}{\partial W_O} = \eta \bar{B} \tag{140}$$

At initialization $W_O = 0$, so by Lemma D.8 the upstream gradient from layer $L$ is

$$G_i^{(L)} = R_i W_O^\top = 0 \tag{141}$$

Using the recurrence (Lemma D.8),

$$G_i^{(l-1)} = G_i^{(l)} + A_i^{(l)\top} G_i^{(l)} V^{(l)\top} + S_i^{(l)} h_i^{(l-1)} W^{(l)\top} + S_i^{(l)\top} h_i^{(l-1)} W^{(l)} \tag{142}$$

Since $G_i^{(L)} = 0$ and $W^{(l)}, V^{(l)} = 0$ for all $l$, we inductively get $G_i^{(l)} = 0$ for every $l$.

Now, the layerwise gradients are (Lemma D.8)

$$\frac{\partial \mathcal{L}}{\partial V^{(l)}} = \frac{-1}{NT} \sum_{i=1}^{N} h_i^{(l-1)\top} A_i^{(l)\top} G_i^{(l)} = 0 \tag{143}$$

and with $S_i^{(l)} = \operatorname{ein}_{tjk,\, tk \to tj} \left( J_i^{(l)}, G_i^{(l)} V^{(l)\top} h_i^{(l-1)\top} \right)$ we also have $S_i^{(l)} = 0$ (because $G_i^{(l)} = 0$ or $V^{(l,0)} = 0$), hence

$$\frac{\partial \mathcal{L}}{\partial W^{(l)}} = \frac{-1}{NT} \sum_{i=1}^{N} h_i^{(l-1)\top} S_i^{(l)} h_i^{(l-1)} = 0 \tag{144}$$

$$\frac{\partial \mathcal{L}}{\partial P^{(l)}} = \frac{-1}{NT} \operatorname{ein}_{tjk,\, jk \to t} \left( D, \sum_{i=1}^{N} S_i^{(l)} \right) = 0 \tag{145}$$

Therefore a single gradient step leaves $W^{(l)}, V^{(l)}, P^{(l)} = 0$ for $1 \leq l \leq L$. $\qquad\square$

**Theorem D.10** (Early Stage Features, Multi-Layer). *Fix a depth $L \leq \frac{\sqrt{T}}{4}$ and assume zero initialization for all parameters. Under the setting described, for $s \leq \eta^{-1} \min\left(\frac{1}{12L}, \frac{5}{8\sqrt{T}}\right)$ with $T \geq 60$ and $|V| \geq 500$, after $s$ gradient descent steps with learning rate $\eta$ we have,* uniformly for every layer $1 \leq l \leq L$,

$$\left\|W_O - s\eta\bar{B}\right\|_F \leq 3s^2\eta^2 \tag{146}$$

$$\left\|V^{(l)} - \binom{s}{2}\eta^2\bar{\Phi}^\top\bar{B}^\top\right\|_F \leq 12s^3\eta^3 \tag{147}$$

$$\left\|W^{(l)} - \left(3\binom{s}{4} + 2\binom{s}{3}\right)\eta^4\bar{Q}\right\|_F \leq 13s^5\eta^5 T \tag{148}$$

$$\left\|P^{(l)} - \left(3\binom{s}{4} + 2\binom{s}{3}\right)\eta^4\Delta\right\|_F \leq 13s^5\eta^5 T \tag{149}$$

*where $\bar{B}$, $\bar{\Phi}$, $\bar{Q}$, and $\Delta$ are as in the one-layer analysis (row-centered bigram matrix, centered prefix-statistics operator, and the third-step structures, respectively).*

*Proof.* We prove the bounds simultaneously for all layers with induction.

By the previous lemma, with zero initialization and one step, $W_O = \eta\bar{B}, W^{(l)} = 0, V^{(l)} = 0, P^{(l)} = 0$ for $1 \leq l \leq L$. This gives the base case.

Now we prove the inductive step. Assume the four bounds hold after $s$ steps, with $(s+1)\eta \leq \min\left(\frac{1}{12L}, \frac{5}{8\sqrt{T}}\right)$. We derive bounds on the deviations in the attention patterns, activations, and outputs in the forward pass after $s$ steps.

Now, we start with a bound on the deviations in the activations from $X_i$ at each row. Since, each row of $A_i^{(l)}$ sums to 1, we have that

$$\left\|h_i^{(l)}[t,:] - X_i[t,:]\right\| \leq \left\|h_i^{(l-1)}[t,:] - X_i[t,:]\right\| + \max_{q \leq T}\left\|h_i^{(l-1)}[q,:]\right\|\left\|V^{(l)}\right\|. \tag{150}$$

and

$$\left\|h_i^{(l)}[t,:]\right\| \leq \left(1 + \left\|V^{(l)}\right\|\right)\left\|h_i^{(l-1)}[t,:]\right\| \tag{151}$$

Then, by the inductive hypothesis and $s\eta \leq \frac{1}{12L}$, we have that $\left\|V^{(l)}\right\|_F \leq 2s^2\eta^2$. Using this and that $h_i^{(0)} = X_i$ which has unit norm, we have that across all layers and rows

$$\left\|h_i^{(l)}[t,:]\right\| \leq \left(1 + 2s^2\eta^2\right)^L \tag{152}$$

Since $s\eta \leq 1/(12L)$, we have

$$2Ls^2\eta^2 \leq \frac{s\eta}{6}.$$

Thus

$$\left(1 + 2s^2\eta^2\right)^L \leq \exp(2Ls^2\eta^2) \leq 1 + \frac{s\eta}{4},$$

where the last inequality uses $2Ls^2\eta^2 \leq 1/72$. Therefore

$$\left\|h_i^{(l)}[t,:]\right\| \leq 1 + \frac{s\eta}{4}.$$

Using this and again that $h_i^{(0)} = X_i$, we have that for all rows and layers,

$$\left\|h_i^{(l)}[t,:] - X_i[t,:]\right\| \leq L\left(1 + \frac{s\eta}{4}\right)2s^2\eta^2 \leq \frac{s\eta}{4},$$

again using $s\eta \leq 1/(12L)$.

Let $A_0$ be the uniform causal attention with the $t$th row having the first $t$ elements equal to $1/t$ and the remaining elements equal to 0. For each layer $l$ and row $t$, define the unmasked attention-logit vector

$$z_{i,t}^{(l)} := \left(h_i^{(l-1)}[t,:]W^{(l)}h_i^{(l-1)\top} + \mathrm{DM}(P^{(l)})[t,:]\right)[:t].$$

Then

$$A_i^{(l)}[t,:t] = \mathcal{S}(z_{i,t}^{(l)}), \qquad A_0[t,:t] = \frac{1}{t}\mathbf{1}.$$

Decomposing $h_i^{(l-1)}[t,:]$ as $X_i[t,:] + (h_i^{(l-1)}[t,:] - X_i[t,:])$, we get by the inductive hypothesis and that $\max_{km} |\tilde{Q}_{km}|, \max_m |\Delta_m| \leq 1$ as shown in the one-layer case that

$$
\begin{aligned}
&\left\| \mathrm{MASK}(h_i^{(l-1)}[t,:]W^{(l)}h_i^{(l-1)\top} + \mathrm{DM}(P^{(l)})[t,:]) \right\| \\
&\leq \left( 6\binom{s}{4} + 4\binom{s}{3} \right) \eta^4\sqrt{t} + 26s^5\eta^5 T \\
&\quad + 2\left\| h_i^{(l-1)}[t,:] - X_i[t,:] \right\| \left( 6\binom{s}{4} + 4\binom{s}{3} \right) \eta^4\sqrt{T} \\
&\quad + \left\| h_i^{(l-1)}[t,:] - X_i[t,:] \right\|^2 \left( 6\binom{s}{4} + 4\binom{s}{3} \right) \eta^4\sqrt{T}
\end{aligned}
\tag{153}
$$

By our earlier bounds, we have then

$$
\begin{aligned}
&\left\| \mathrm{MASK}(h_i^{(l-1)}[t,:]W^{(l)}h_i^{(l-1)\top} + \mathrm{DM}(P^{(l)})[t,:]) \right\| \\
&\leq s^4\eta^4\sqrt{t} + 26s^5\eta^5 T + \frac{s\eta}{2}s^4\eta^4\sqrt{T} + \frac{s^2\eta^2}{16}s^4\eta^4\sqrt{T}
\end{aligned}
\tag{154}
$$

which we can upper bound by

$$\left\| \mathrm{MASK}(h_i^{(l-1)}[t,:]W^{(l)}h_i^{(l-1)\top} + \mathrm{DM}(P^{(l)})[t,:]) \right\| \leq s^4\eta^4\sqrt{t} + \frac{21}{2}s^3\eta^3 \tag{155}$$

and we have

$$\left\| z_{i,t}^{(l)} \right\| \leq s^4\eta^4\sqrt{t} + \frac{21}{2}s^3\eta^3. \tag{156}$$

Moreover, since $\|z_{i,t}^{(l)}\|_\infty \leq \|z_{i,t}^{(l)}\|$, the preceding bound and the assumptions $s\eta \leq 5/(8\sqrt{T})$ and $T \geq 60$ imply

$$\|z_{i,t}^{(l)}\|_\infty \leq \frac{1}{2}.$$

Hence

$$\mathrm{range}(z_{i,t}^{(l)}) \leq 2\|z_{i,t}^{(l)}\|_\infty \leq 1.$$

By Lemma D.2, applied to the softmax on the first $t$ coordinates,

$$\left\| (A_i^{(l)} - A_0)[t,:] \right\| \leq \frac{e}{t}\left\| z_{i,t}^{(l)} \right\|.$$

Therefore

$$\left\| (A_i^{(l)} - A_0)[t,:] \right\| \leq \frac{es^4\eta^4}{\sqrt{t}} + \frac{21e}{2}\frac{s^3\eta^3}{t}.$$

For $t = 1$, both $A_i^{(l)}[1,:]$ and $A_0[1,:]$ are point masses, so the left-hand side is zero. For $t \geq 2$, the assumptions $s\eta \leq 5/(8\sqrt{T})$ and $T \geq 60$ imply

$$\frac{es^4\eta^4}{\sqrt{t}} + \frac{21e}{2}\frac{s^3\eta^3}{t} \leq \frac{s\eta}{\sqrt{T}}.$$

Thus, for all $t$,

$$\left\| (A_i^{(l)} - A_0)[t,:] \right\| \leq \frac{s\eta}{\sqrt{T}}. \tag{157}$$

Consequently,

$$\left\| A_i^{(l)} - A_0 \right\|_F \leq s\eta. \tag{158}$$

From the deviation bounds on the activations and the inductive control of $W_O$,

$$\|F_\theta(X_i)\|_F = \|h_i^{(L)}W_O\|_F \leq \|h_i^{(L)}\|_F\|W_O\| \leq (1 + \frac{s\eta}{4})\sqrt{T}(s\eta + 3s^2\eta^2) \leq 2s\eta\sqrt{T} \tag{159}$$

For each row $t$, let
$$f_{i,t} := F_\theta(X_i)^{[t,:]} = h_i^{(L)}[t,:]W_O.$$
Using the activation bound and the inductive hypothesis for $W_O$,
$$\|f_{i,t}\| \le \left(1 + \frac{s\eta}{4}\right)(s\eta + 3s^2\eta^2).$$
Since $s\eta \le 5/(8\sqrt{T})$ and $T \ge 60$, the right-hand side is at most 1. Hence
$$\text{range}(f_{i,t}) \le 2\|f_{i,t}\| \le 2.$$
By Lemma D.2, applied row-wise,
$$\left\|\mathcal{S}(f_{i,t}) - \frac{1}{|V|}\mathbf{1}\right\| \le \frac{e^2}{|V|}\|f_{i,t}\|.$$
Summing over rows gives
$$\|\mathcal{S}(F_\theta(X_i)) - U_O\|_F \le \frac{e^2}{|V|}\|F_\theta(X_i)\|_F \le \frac{2e^2 s\eta\sqrt{T}}{|V|}.$$
Since $|V| \ge 500$, we have $e^2/\sqrt{|V|} \le 1$, and therefore
$$\|\mathcal{S}(F_\theta(X_i)) - U_O\|_F \le 2s\eta\sqrt{\frac{T}{|V|}}. \tag{160}$$

Then as in the one-layer case but using equation 160 and accounting for deviations in the hidden state from $X_i$,
$$\left\|\frac{\partial\mathcal{L}}{\partial W_O} + \bar{B}\right\|_F \le \frac{4s\eta}{\sqrt{|V|}} + 4s\eta \le 5s\eta \tag{161}$$
Then, after $s+1$ steps, we have
$$\left\|W_O - (s+1)\eta\bar{B}\right\|_F \le 3s^2\eta^2 + 5s\eta^2 \le 3(s+1)^2\eta^2 \tag{162}$$
From Lemma D.8,
$$\frac{\partial\mathcal{L}}{\partial V^{(l)}} = -\frac{1}{NT}\sum_i h_i^{(l-1)\top} A_i^{(l)\top} R_i W_O^\top \tag{163}$$
Considering the deviation from each of the terms, we have
$$\left\|\frac{\partial\mathcal{L}}{\partial V^{(l)}} + s\eta\bar{\Phi}^\top\bar{B}^\top\right\|_F \le 36s^2\eta^2 \tag{164}$$
Then, we have that after $s+1$ steps,
$$\left\|V^{(l)} - \binom{s+1}{2}\eta^2\bar{\Phi}^\top\bar{B}^\top\right\|_F \le 12s^3\eta^3 + 36s^2\eta^3 \le 12(s+1)^3\eta^3$$
From Lemma D.8,
$$\frac{\partial\mathcal{L}}{\partial W^{(l)}} = -\frac{1}{NT}\sum_i h_i^{(l-1)\top} S_i^{(l)} h_i^{(l-1)} \tag{165}$$
$$\frac{\partial\mathcal{L}}{\partial P^{(l)}} = -\frac{1}{NT}\text{ein}_{tjk,\, jk\to t}\left(D, \sum_i S_i^{(l)}\right) \tag{166}$$
with $S_i^{(l)} = \text{ein}\left(J_i^{(l)}, G_i^{(l)} V^{(l)\top} h_i^{(l-1)\top}\right)$. As in the one-layer bound, we can use the bound on the attention pattern to control $J_i^{(l)}$. We have that for $t \ge 2$
$$\|J_{t,i} - J_t\|_2 \le \left(1 + \frac{2}{\sqrt{t}}\right)\|A_i[t,:] - A_0[t,:]\|_2 + \|A_i[t,:] - A_0[t,:]\|_2^2 \tag{167}$$

Then, as we have that

$$\|A_i[t,:] - A_0[t,:]\|_2 \leq \frac{s\eta}{\sqrt{T}} \tag{168}$$

it follows that

$$\|J_{t,i} - J_t\|_2 \leq \frac{10s\eta}{\sqrt{T}} \tag{169}$$

Since $J_{1,i}$ is always all zeros, we can ignore this term and for $t \geq 2$, we have that as $s\eta \leq \frac{5}{8\sqrt{T}}$,

$$\|J_{t,i} - J_t\|_2 \leq \frac{10s\eta}{\sqrt{T}} \tag{170}$$

Now, we bound the deviation of $G_i^{(l)}$ from $s\eta(Y_i - U_O)\bar{B}^\top$. Starting from layer $L$, we have

$$\left\|G_i^{(L)} - s\eta(Y_i - U_O)\bar{B}^\top\right\|_F \leq 2s\eta\sqrt{\frac{T}{|V|}}(2s\eta) + \sqrt{T}(3s^2\eta^2) \leq 4s^2\eta^2\sqrt{T} \tag{171}$$

We will let the bound on the deviation at layer $l$ be $D_{G,l}$. Now, we consider the bound for each layer $l$,

$$\left\|G_i^{(l-1)} - s\eta(Y_i - U_O)\bar{B}^\top\right\|_F \leq D_{G,l} + 4s^2\eta^2\left\|G_i^{(l)}\right\| + 2\left\|S_i^{(l)}\right\|(2\sqrt{T})(2s^4\eta^4T) \tag{172}$$

Since we also need the norm of $S_i^{(l)}$ to iterate through layers, we bound the norm of $S_i^{(l)}$,

$$\begin{aligned}
\left\|S_i^{(l)}\right\|_F &\leq \left\|J_i^{(l)}\right\|\left\|G_i^{(l)}\right\|\left\|V^{(l)}\right\|\left\|h_i^{(l-1)}\right\|_F \\
&\leq \frac{5}{2}\left\|G_i^{(l)}\right\|(2s^2\eta^2)(2\sqrt{T}) \\
&\leq 10s^2\eta^2\left\|G_i^{(l)}\right\|\sqrt{T} \\
&\leq 7s\eta\left\|G_i^{(l)}\right\|,
\end{aligned} \tag{173}$$

Using this upper bound back in the recurrence for $D_{G,l}$, we have

$$\left\|G_i^{(l-1)} - s\eta(Y_i - U_O)\bar{B}^\top\right\|_F \leq D_{G,l} + 4s^2\eta^2\left\|G_i^{(l)}\right\| + 56s^5\eta^5T^{3/2}\left\|G_i^{(l)}\right\| \tag{174}$$

Using $s\eta \leq \frac{5}{8\sqrt{T}}$, we have

$$\left\|G_i^{(l-1)} - s\eta(Y_i - U_O)\bar{B}^\top\right\|_F \leq D_{G,l} + 18s^2\eta^2\left\|G_i^{(l)}\right\| \tag{175}$$

We can now write a recurrence for $\left\|G_i^{(l)}\right\|$ as $\left\|G_i^{(l)}\right\| \leq s\eta\left\|(Y_i - U_O)\bar{B}^\top\right\| + D_{G,l}$, we have

$$\left\|G_i^{(l-1)}\right\| \leq (1 + 18s^2\eta^2)(\left\|s\eta(Y_i - U_O)\bar{B}^\top\right\| + D_{G,l}) \leq (1 + 18s^2\eta^2)(s\eta\sqrt{2T} + D_{G,l}) \tag{176}$$

Utilizing this with the recurrence for $D_{G,l}$, we can then write a recurrence only in terms of $D_{G,l}$ and find that for all $l$, $D_{G,l} \leq 12s^2\eta^2\sqrt{T}$ as $L \leq \frac{\sqrt{T}}{4}$ and $s\eta \leq \min\left(\frac{1}{12L}, \frac{5}{8\sqrt{T}}\right)$. Then, we also have that for all $l$, $\left\|G_i^{(l)}\right\| \leq s\eta\sqrt{T} + 12s^2\eta^2\sqrt{T} \leq 2s\eta\sqrt{T}$. Then, we have that as $\|J_t\|_2 = \frac{1}{t}$, $\|J_i\|_2 \leq \frac{3}{2} + \frac{10s\eta}{\sqrt{T}} \leq 2$, and $s\eta \leq \min\left(\frac{1}{12L}, \frac{5}{8\sqrt{T}}\right)$,

$$\left\|\frac{s^3 - s^2}{2}\eta^3Q_i - S_i^{(l)}\right\|_F \leq 64s^4\eta^4T \tag{177}$$

This produces

$$\left\|\frac{\partial\mathcal{L}}{\partial W^{(l)}} + \frac{s^3 - s^2}{2}\eta^3\bar{Q}\right\|_F \leq 64s^4\eta^4T \tag{178}$$

and similarly,

$$\left\|\frac{\partial \mathcal{L}}{\partial P^{(l)}} + \frac{s^3 - s^2}{2}\eta^3\Delta\right\|_F \le 64s^4\eta^4 T \tag{179}$$

and hence after $(s + 1)$ steps

$$\left\|W^{(l)} - \left(3\binom{s+1}{4} + 2\binom{s+1}{3}\right)\eta^4\bar{Q}\right\|_F \le 13(s+1)^5\eta^5 T \tag{180}$$

$$\left\|P^{(l)} - \left(3\binom{s+1}{4} + 2\binom{s+1}{3}\right)\eta^4\Delta\right\|_F \le 13(s+1)^5\eta^5 T \tag{181}$$

$\square$

**Lemma D.11** (Gaussian Initialization Operator Norm). *Under the setting described and with all parameters initialized from $\mathcal{N}(0, \frac{v^2}{|V|^{2+2\xi}})$ for $\xi \ge 0$ and $T \le |V|$, we have that with probability at least $1 - (3L + 1)\exp\left(-\frac{|V|^{1+2\xi}}{4}\right)$, for all $1 \le l \le L$,*

$$\|W_O\|, \left\|V^{(l)}\right\|, \left\|W^{(l)}\right\|, \left\|P^{(l)}\right\| \le \frac{3v}{|V|^{1/2}} \tag{182}$$

*Proof.* We start with $W_O$. Using a concentration bound on Gaussian random matrices, we have that

$$\mathbb{P}\left(\left\|\frac{|V|^{1+\xi}}{v}W_O\right\| \ge 2\sqrt{|V|} + t\right) \le e^{-t^2/2} \tag{183}$$

Then, setting $t = |V|^{1/2+\xi}$, we have that

$$\mathbb{P}\left(\|W_O\| \ge \frac{3v}{|V|^{1/2}}\right) \le \exp\left(-\frac{|V|^{1+2\xi}}{2}\right) \tag{184}$$

Then, with probability at least $1 - \exp\left(-\frac{|V|^{1+2\xi}}{2}\right)$,

$$\|W_O\| \le \frac{3v}{|V|^{1/2}} \tag{185}$$

We can apply the same argument for each of $V^{(l)}, W^{(l)}$ to derive the same bound. Since $P^{(l)}$ is smaller than the other matrices and has the same initialization, we can also apply the same bound. Applying a union bound on the probability of failures for each of the weights, we have that with probability at least $1 - (3L + 1)\exp\left(-\frac{|V|^{1+2\xi}}{2}\right)$, all of $W_O, V^{(l)}, W^{(l)}, P^{(l)}$ have operator norm at most $\frac{3v}{|V|^{1/2}}$. $\square$

**Lemma D.12** (Gaussian Initialization Frobenius Norm). *Under the setting described and with all parameters initialized from $\mathcal{N}(0, \frac{v^2}{|V|^{2+2\xi}})$ for $\xi \ge 0$ and $T \le |V|$, we have that with probability at least $1 - (3L + 1)\exp\left(-\frac{|V|^{2+2\xi}}{4}\right)$, for all $1 \le l \le L$,*

$$\|W_O\|_F, \left\|V^{(l)}\right\|_F, \left\|W^{(l)}\right\|_F, \left\|P^{(l)}\right\|_F \le 2v \tag{186}$$

*Proof.* We start with $W_O$. Using Lemma 1 from Laurent & Massart (2000), we have that

$$\mathbb{P}\left(\|W_O\|_F^2 \ge \frac{v^2}{|V|^{2+2\xi}}\left(|V|^2 + 2|V|\sqrt{t} + 2t\right)\right) \le e^{-t} \tag{187}$$

Then, setting $t = \frac{|V|^{2+2\xi}}{4}$, we have that

$$\mathbb{P}\left(\|W_O\|_F^2 \ge 3v^2\right) \le \exp\left(-\frac{|V|^{2+2\xi}}{4}\right) \tag{188}$$

Then, with probability at least $1 - \exp\left(-\frac{|V|^{2+2\xi}}{4}\right)$,

$$\|W_O\|_F \leq 2v \tag{189}$$

We can apply the same argument for each of $V^{(l)}, W^{(l)}$ to derive the same bound. For $P^{(l)}$, we have

$$\mathbb{P}\left(\left\|P^{(1)}\right\|_F^2 \geq \frac{v^2}{|V|^{2+2\xi}}(T + 2\sqrt{Tt} + 2t)\right) \leq e^{-t} \tag{190}$$

Then, setting $t = \frac{|V|^{2+2\xi}}{4}$ and using that $T \leq |V|$, we have that

$$\mathbb{P}\left(\left\|P^{(1)}\right\|_F^2 \geq 3v^2\right) \leq \exp\left(-\frac{|V|^{2+2\xi}}{4}\right) \tag{191}$$

Then, with probability at least $1 - \exp\left(-\frac{|V|^{2+2\xi}}{4}\right)$,

$$\left\|P^{(1)}\right\|_F \leq 2v \tag{192}$$

Applying a union bound on the probability of failures for each of the weights, we have that with probability at least $1 - (3L + 1)\exp\left(-\frac{|V|^{2+2\xi}}{4}\right)$, all of $W_O, V^{(l)}, W^{(l)}, P^{(l)}$ have Frobenius norm at most $2v$. $\qquad\square$

**Theorem D.13** (Gaussian Initialization (Multi-Layer)). *Assume the setting of D.10 with depth $L \leq \frac{\sqrt{T}}{4}$, all parameters initialized i.i.d. from $\mathcal{N}\left(0, \frac{v^2}{|V|^{2+2\xi}}\right)$ with $v \leq \frac{\eta^2}{T^2}$, $T \leq |V|$, and learning rate $\eta \geq T^{-1}$. Then, with probability at least*

$$1 - (3L + 1)\left[\exp\left(-\frac{|V|^{1+2\xi}}{4}\right) + \exp\left(-\frac{|V|^{2+2\xi}}{4}\right)\right]$$

*for $s \leq \eta^{-1}\min\left(\frac{1}{12L}, \frac{5}{8\sqrt{T}}\right)$ with $T \geq 60$ and $|V| \geq 500$, after $s$ gradient descent steps with learning rate $\eta$ we have,* uniformly *for every layer $1 \leq l \leq L$,*

$$\left\|W_O - s\eta\bar{B}\right\|_F \leq 3s^2\eta^2 \tag{193}$$

$$\left\|V^{(l)} - \binom{s}{2}\eta^2\bar{\Phi}^\top\bar{B}^\top\right\|_F \leq 12s^3\eta^3 \tag{194}$$

$$\left\|W^{(l)} - \left(3\binom{s}{4} + 2\binom{s}{3}\right)\eta^4\bar{Q}\right\|_F \leq 13s^5\eta^5 T \tag{195}$$

$$\left\|P^{(l)} - \left(3\binom{s}{4} + 2\binom{s}{3}\right)\eta^4\Delta\right\|_F \leq 13s^5\eta^5 T \tag{196}$$

*where $\bar{B}, \bar{\Phi}, \bar{Q}$, and $\Delta$ are as in the one-layer analysis (row-centered bigram matrix, centered prefix-statistics operator, and the third-step structures, respectively).*

*Proof.* We start by noting that as long the proof holds when $v = \frac{\eta^2}{T^2}$ and we show that the first gradient step satisfies the inductive hypothesis used in Theorem D.10, then the proof will be complete. We will prove that the first gradient step satisfies the inductive hypothesis with $v = \frac{\eta^2}{T^2}$.

We will condition on the event that the results of Lemmas D.11 and D.12 holds. Then, our results will hold with probability at least

$$1 - (3L + 1)\left[\exp\left(-\frac{|V|^{1+2\xi}}{4}\right) + \exp\left(-\frac{|V|^{2+2\xi}}{4}\right)\right]$$

and we have that at initialization for all $1 \leq l \leq L$

$$\|W_O\|, \left\|V^{(l)}\right\|, \left\|W^{(l)}\right\|, \left\|P^{(l)}\right\| \leq \frac{3\eta^2}{T^2|V|^{1/2}} \tag{197}$$

and

$$\|W_O\|_F, \left\|V^{(l)}\right\|_F, \left\|W^{(l)}\right\|_F, \left\|P^{(l)}\right\|_F \leq \frac{2\eta^2}{T^2} \tag{198}$$

Now, we start with a bound on the deviations in the activations from $X_i$ at each row. Since, each row of $A_i^{(l)}$ sums to 1, we have that

$$\left\|h_i^{(l)}[t, :] - X_i[t, :]\right\| \leq \left\|h_i^{(l-1)}[t, :] - X_i[t, :]\right\| + \max_{q \leq T} \left\|h_i^{(l-1)}[q, :]\right\| \left\|V^{(l)}\right\|. \tag{199}$$

and

$$\left\|h_i^{(l)}[t, :]\right\| \leq (1 + \left\|V^{(l)}\right\|) \left\|h_i^{(l-1)}[t, :]\right\| \tag{200}$$

Using that $\left\|V^{(l)}\right\| \leq \frac{3\eta^2}{T^2|V|^{1/2}}$ and that $h_i^{(0)} = X_i$ which has unit norm, we have that across all layers and rows

$$\left\|h_i^{(l)}[t, :]\right\| \leq \left(1 + \frac{3\eta^2}{T^2|V|^{1/2}}\right)^L \tag{201}$$

and as $\eta \leq \frac{1}{12L}$ and as $(1 + c/L)^L \leq 1 + 2c$ for $c \leq 1$, we have that

$$\left\|h_i^{(l)}[t, :]\right\| \leq 1 + \frac{\eta^{7/2}}{2} \tag{202}$$

Using this and again that $h_i^{(0)} = X_i$, we have that for all rows and layers,

$$\left\|h_i^{(l)}[t, :] - X_i[t, :]\right\| \leq L\left(1 + \frac{\eta^{7/2}}{2}\right)\frac{3\eta^2}{T^2|V|^{1/2}} \leq \frac{\eta^{7/2}}{3} \tag{203}$$

again using that $s\eta \leq \frac{1}{12L}$.

Let $A_0$ be the uniform causal attention with the $t$-th row having the first $t$ elements equal to $1/t$ and the remaining elements being 0. By our earlier bounds, we have then

$$\begin{aligned}
&\left\|\text{MASK}(h_i^{(l-1)}[t, :]W^{(l)}h_i^{(l-1)\top} + \text{DM}(P^{(l)})[t, :])\right\| \\
&\leq \left(1 + \frac{\eta^{7/2}}{2}\right)^2 \frac{3\eta^2}{T^2|V|^{1/2}}\sqrt{T} + \frac{3\eta^2}{T^2|V|^{1/2}} \leq \frac{6\eta^{7/2}}{\sqrt{T}}
\end{aligned} \tag{204}$$

where we have use $\frac{1}{T} \leq \eta$ and $\eta \leq \frac{1}{12L}$. For the $t$th row, define

$$z_{i,t}^{(l)} := \left(h_i^{(l-1)}[t, :]W^{(l)}h_i^{(l-1)\top} + \text{DM}(P^{(l)})[t, :]\right)[: t].$$

The preceding bound gives

$$\left\|z_{i,t}^{(l)}\right\| \leq \frac{6\eta^{7/2}}{\sqrt{T}}.$$

In particular,

$$\left\|z_{i,t}^{(l)}\right\|_\infty \leq \frac{6\eta^{7/2}}{\sqrt{T}} \leq \frac{1}{2},$$

under the assumptions on $\eta$ and $T$. Hence

$$\text{range}(z_{i,t}^{(l)}) \leq 1.$$

By Lemma D.2, applied to the softmax on the first $t$ coordinates,

$$\left\|(A_i^{(l)} - A_0)[t, :]\right\| \leq \frac{e}{t}\left\|z_{i,t}^{(l)}\right\| \leq \frac{6e\eta^{7/2}}{\sqrt{T}}.$$

Consequently,

$$\left\|A_i^{(l)} - A_0\right\|_F \leq 6e\eta^{7/2}.$$

From the deviation bounds on the activations and the initial bound on $W_O$,

$$\|F_\theta(X_i)\|_F = \|h_i^{(L)} W_O\|_F \le \|h_i^{(L)}\|_F \|W_O\| \le (1 + \frac{\eta^{7/2}}{2})\sqrt{T}\frac{3\eta^2}{T^2|V|^{1/2}} \le 4\eta^4 \qquad (205)$$

For each row $t$, let

$$f_{i,t} = F_\theta(X_i)^{[t,:]}.$$

From the preceding bound,

$$\|f_{i,t}\| \le \|F_\theta(X_i)\|_F \le 4\eta^4.$$

Thus, under the smallness assumptions on $\eta$,

$$\mathrm{range}(f_{i,t}) \le 2\|f_{i,t}\| \le 1.$$

By Lemma D.2, applied row-wise,

$$\left\| \mathcal{S}(f_{i,t}) - \frac{1}{|V|}\mathbf{1} \right\| \le \frac{e}{|V|}\|f_{i,t}\|.$$

Summing over rows gives

$$\|\mathcal{S}(F_\theta(X_i)) - U_O\|_F \le \frac{e}{|V|}\|F_\theta(X_i)\|_F \le \frac{4e\eta^4}{|V|} \le \frac{4\eta^4}{\sqrt{|V|}},$$

where the last inequality uses $|V| \ge 500$.

Then following the argument in the zero-initialization case,

$$\left\| \frac{\partial \mathcal{L}}{\partial W_O} + \bar{B} \right\|_F \le \frac{4\eta^4}{\sqrt{|V|}} + \frac{\eta^{5/2}}{\sqrt{T}} \le 2\eta^3 \qquad (206)$$

Then, after the first step,

$$\left\| W_O - \eta\bar{B} \right\|_F \le \frac{3\eta^2}{T^2|V|^{1/2}} + 2\eta^4 \le 3\eta^4 \le 3\eta^2 \qquad (207)$$

From Lemma D.8,

$$\frac{\partial \mathcal{L}}{\partial V^{(l)}} = -\frac{1}{NT}\sum_i h_i^{(l-1)\top} A_i^{(l)\top} R_i W_O^\top \qquad (208)$$

Considering the deviation from each of the terms, we have

$$\left\| \frac{\partial \mathcal{L}}{\partial V^{(l)}} \right\|_F \le \frac{15\eta^2}{T^2|V|^{1/2}} \le 15\eta^{9/2} \qquad (209)$$

Then, we have that after the first step

$$\left\| V^{(l)} \right\|_F \le \frac{2\eta^2}{T^2} + 15\eta^{11/2} \le 3\eta^4 \le 12\eta^3$$

From Lemma D.8,

$$\frac{\partial \mathcal{L}}{\partial W^{(l)}} = -\frac{1}{NT}\sum_i h_i^{(l-1)\top} S_i^{(l)} h_i^{(l-1)} \qquad (210)$$

$$\frac{\partial \mathcal{L}}{\partial P^{(l)}} = -\frac{1}{NT}\mathrm{ein}_{tjk,jk\to t}\Big(D, \sum_i S_i^{(l)}\Big) \qquad (211)$$

with $S_i^{(l)} = \mathrm{ein}\big(J_i^{(l)}, G_i^{(l)} V^{(l)\top} h_i^{(l-1)\top}\big)$. As in the zero-initialization case, we can use the bound on the attention pattern to control $J_i^{(l)}$. We have that for $t \ge 2$

$$\|J_{t,i} - J_t\|_2 \le \left(1 + \frac{2}{\sqrt{t}}\right)\|A_i[t,:] - A_0[t,:]\|_2 + \|A_i[t,:] - A_0[t,:]\|_2^2 \qquad (212)$$

Then, as we have that

$$\|A_i[t,:] - A_0[t,:]\|_2 \leq \frac{6e\eta^{7/2}}{\sqrt{T}} \tag{213}$$

it follows that

$$\|J_{t,i} - J_t\|_2 \leq \frac{50\eta^{7/2}}{\sqrt{T}} \tag{214}$$

Since $J_{1,i}$ is always all zeros, we can ignore this term and for $t \geq 2$, we have that,

$$\|J_i - J\|_2 \leq 50\eta^{7/2} \tag{215}$$

Now, we bound the norm of $G_i^{(l)}$. Starting from layer $L$, we have

$$\left\|G_i^{(L)}\right\|_F \leq \sqrt{2T}\frac{3\eta^2}{T^2|V|^{1/2}} \leq 5\eta^4 \tag{216}$$

Let $D_{G,l}$ denote a uniform upper bound on $\left\|G_i^{(l)}\right\|_F$. Now, we consider the bound for each layer $l$,

$$\left\|G_i^{(l-1)}\right\|_F \leq D_{G,l} + \frac{5}{2}D_{G,l}\frac{3\eta^2}{T^2|V|^{1/2}} + 2\left\|S_i^{(l)}\right\|\sqrt{2T}\frac{3\eta^2}{T^2|V|^{1/2}} \leq (1+8\eta^{9/2})D_{G,l} + 9\eta^4\left\|S_i^{(l)}\right\| \tag{217}$$

Since we also need the norm of $S_i^{(l)}$ to iterate through layers, we bound the norm of $S_i^{(l)}$,

$$\left\|S_i^{(l)}\right\|_F \leq \left\|J_i^{(l)}\right\|\left\|G_i^{(l)}\right\|\left\|V^{(l)}\right\|\left\|h_i^{(l-1)}\right\|_F \leq \frac{5}{2}D_{G,l}\frac{3\eta^2}{T^2|V|^{1/2}}\sqrt{2T} \leq 8\eta^4 D_{G,l} \tag{218}$$

Using this upper bound back in the recurrence for $D_{G,l}$, we have

$$\left\|G_i^{(l-1)}\right\|_F \leq (1+8\eta^{9/2}+72\eta^8)D_{G,l} \tag{219}$$

Then for all $l$, $D_{G,l} \leq 6\eta^4$ as $L \leq \frac{\sqrt{T}}{4}$ and $\eta \leq \min\left(\frac{1}{12L}, \frac{5}{8\sqrt{T}}\right)$. Then, we also have that for all $l$, $\left\|G_i^{(l)}\right\| \leq 6\eta^4$. Then, we have that as $\|J_t\|_2 = \frac{1}{t}$, $\left\|J_i^{(l)}\right\|_2 \leq \frac{3}{2} + 50\eta^{7/2} \leq 2$, and $s\eta \leq \min\left(\frac{1}{12L}, \frac{5}{8\sqrt{T}}\right)$,

$$\left\|S_i^{(l)}\right\|_F \leq 48\eta^8 \tag{220}$$

This produces

$$\left\|\frac{\partial\mathcal{L}}{\partial W^{(l)}}\right\|_F \leq 48\eta^8 \tag{221}$$

and similarly,

$$\left\|\frac{\partial\mathcal{L}}{\partial P^{(l)}}\right\|_F \leq 48\eta^8 \tag{222}$$

and hence after the first step

$$\left\|W^{(l)}\right\|_F \leq 48\eta^9 + \frac{3\eta^2}{T^2} \leq 4\eta^5 T$$

$$\left\|P^{(l)}\right\|_F \leq 48\eta^9 + \frac{3\eta^2}{T^2} \leq 4\eta^5 T$$

$\square$

