# OpenReview forum: "How Do Transformers Learn to Associate Tokens: Gradient Leading Terms Bring Mechanistic Interpretability"
_ICLR.cc/2026/Conference — ICLR 2026 Oral_

### Official Review · Reviewer_XVTW · 2025-10-15

**Soundness:** 3
**Presentation:** 3
**Contribution:** 4
**Rating:** 8
**Confidence:** 4

**Summary:**

This paper study how transformers learn semantic associations during training on natural language data. The authors develop a theory based on training dynamics, specifically using a leading-term approximation of the gradients to analyze the model's weights in the early phase of training. Their central finding is that the weight matrices, like the output $W_O$, the value $V^{(\ell)}$, and query-key $W^{(\ell)}$ can be expressed as compositions of three basis functions derived directly from corpus statistics. These basis functions are bigram maping, that captures the next token co-occurrence ($\bar{B}$); the interchangeability maping, that fnds tokens with similar preceding token distributions (like synonyms or words in the same grammatical class) ($\sum_{\bar{B}}$); and context mapping, that encodes long range co-occurrance between a token and the tokens in its prefix ($\bar{\Phi}$).
The authors validate this theory first on a 3 layer attention only model trained on TinyStories, and they show that the learned weights have a very high cosine similarity (often >0.99) to their theoretical characterizations. Then, they extend the analysis to the Pythia-1.4B checkpoints, showing that these foundational associations are also learned in larger models, especially during early training and in the later layers.

**Strengths:**

- The leading-term gradient approximation is a clever technical innovation that makes an otherwise intractable analysis feasible while maintaining interpretability. The decomposition into three basis functions provides clear mechanistic insight into how different components capture semantic structure.
- Unlike much prior theoretical work that relies on synthetic data or heavily simplified models, this analysis is grounded in a more realistic setup.
- The paper provides closed-form expressions for all major weight matrices uniformly across layers, showing how they compose the basis functions differently.

**Weaknesses:**

- The theory applies to attention-only transformers, but validation on Pythia requires indirect analysis through covariance matrices of activations rather than direct weight comparison. This introduces an additional layer of approximation and makes it unclear how MLPs and multi-head attention affect the theoretical predictions.
- The theoretical guarantees only hold for $\mathcal{O}(1/\eta)$ steps in early training. While empirical results show features persist longer, the paper lacks analysis of how and why these features evolve during later training, limiting practical applicability for understanding fully trained models.
- To achieve theoretical tractability, the analysis relies on certain assumptions, such as "sufficiently small" initialization and bounds on the learning rate and number of steps (as seen in Theorem 4.1). The error bounds for the approximation, while derived, grow over time. This means the exact closed-form expressions are an idealization, and their accuracy naturally decreases as training progresses and the model's state becomes more complex.
- Experiments use truncated vocabularies (3K for TinyStories, 20K for Pythia) which may not capture the full complexity of semantic associations in real LLMs with vocabularies of 50K+ tokens. The computational tractability argument doesn't fully justify this limitation.

**Questions:**

Your leading-term approximation brilliantly explains how associations emerge in the very early stages of training. The empirical results with Pythia suggest these structures persist for some time but also evolve, especially in the earlier layers. What are your thoughts on the mechanisms that drive this evolution away from the initial basis functions? Is it simply the accumulation of higher-order gradient terms, or do you suspect qualitatively new structures, like logical reasoning circuits, are being built on top of this initial associative foundation?

---

> ### Author Response · Authors · 2025-11-20
>
> We sincerely appreciate your positive feedback and insightful comments! We address the questions and comments below in detail.
>
>
> > `W1`. Applicability to the transformers with MLPs and multi-head attention.
>
> We appreciate your insightful comment here and address both aspects below.
>
> **(1) MLP.** We fully agree that understanding the impact of MLP layers is important. In response to this comment, we conducted additional analysis on how MLP layers interact with the basis functions. The high complexity of MLP transformations makes deriving clean theoretical statements challenging, especially across different checkpoints across training steps. Nevertheless, our empirical results on Pythia indicate that embeddings with both the value/output mapping and the MLP output added **still preserve the token-level structure predicted by our analysis**. Our additional analysis (**Section 5.2**) **compares embeddings with and without the MLP output, and we find they capture similar token-associations** with cosine similarity scores differing only by 0.01-0.1. This demonstrates that the theoretical basis features persist despite the added architectural complexity and remain meaningful in full transformer models.
>
> **(2) Multi-head attention**. Similarly, theoretically analyzing the dynamics of multi-head attention is challenging due to its complexity, so we have provided additional experiments looking into individual heads. In our new experiments (**Figure 7**), we explored the behavior of specific attention heads, providing a comparison of how quickly heads drift from our leading term expression over time. We find that
>
> 1. Earlier layers learn the leading-term features more slowly
> 2. Intermediate layers seem to have individual heads specialize faster than other layers.
>
> > `W2`. Analysis and guarantees for later stage behavior
>
> We agree with your comment that being able to analyze later stages of training would be insightful. However, we believe this limitation is appropriate for the scope of our contribution, for two key reasons:
>
> 1. The goal of the paper is to explain **how the _core, foundational_ associative features emerge**, not to fully characterize all representational refinements.
>     The early-training phase is where the model first builds the foundational token-association structure (e.g., bigram-like associations, context basis). Our theorem captures this mechanistic process with a clean, realistic analysis, as you recognized.
>
> 2. Later-stage training requires addressing multiple fundamental problems, such as higher-order gradient interactions, semantic specialization, and cross-layer coupling. The challenges, along with the various phenomena seen in models at later stages, suggest that providing a general characterization of model features at these stages will require careful analysis on its own and new tools beyond gradient leading term. _Attempting to cover both regimes in a single paper would produce a much less focused and less rigorous contribution_. Nonetheless, our insights on how semantic relationships start to form can inform hypotheses on how these develop into more complex capabilities. We believe these insights can guide both theorists and practitioners to invent a follow-up theory and algorithm for interpretable AI in the more challenging setups.
>
> > `W3`. Deviations between the theoretical characterization and reality over training
>
> We appreciate the comment pointing out the growing deviation between the closed-form expressions and model weights. As our analysis is focused on the early training stage, we do not expect the derived expressions to generalize across the full training trajectory. We would like to emphasize that our analyses and results do not aim to idealize how fully-trained models behave but rather describe the features learned at an early stage to build a theoretically grounded base for future work in interpretability.
>
> > `W4`. Large vocabulary size
>
> We appreciate the feedback on vocabulary size and have updated results to use the full vocabulary (50,304) for Pythia experiments and added results for TinyStories based on a BPE tokenization with 10,000 tokens in **Appendix B**. We find that even with the full vocabulary for Pythia, the results remain similar (Figure 6), and for TinyStories with BPE, we find that the cosine similarity between the theoretical and learned weights **remains above 0.997 for 10 epochs**.

---

> > ### Author Response · Authors · 2025-11-20
> >
> > > `Q1`. Mechanisms of feature evolution over training steps.
> >
> > That's a very inspiring question! The evolution away from the initial basis functions likely corresponds to learning higher-level semantics rather than overfitting, since we know that the model becomes more capable during training at later stages and forms structures such as induction heads and better representations of higher-level concepts [1, 2]. As discussed in `W2`, the various observations about model behavior at later stages suggest that providing a general characterization of model features at these stages will require careful analysis. Whether the new features should be thought of as being derived from higher-order terms or building circuits on top of the initial features is a great question and one that we hope can be explored further.
> >
> > [1] Belrose, Nora, et al. "Neural networks learn statistics of increasing complexity." arXiv preprint arXiv:2402.04362 (2024).
> >
> > [2] Wang, George, et al. "Loss landscape geometry reveals stagewise development of transformers." High-dimensional Learning Dynamics 2024: The Emergence of Structure and Reasoning. 2024.

---

> > > ### Comment · Reviewer_XVTW · 2025-11-27
> > >
> > > I read the authors' response and I appreciated the extra informations they added.
> > > Thank you

---

### Official Review · Reviewer_T8ho · 2025-10-26

**Soundness:** 3
**Presentation:** 3
**Contribution:** 3
**Rating:** 6
**Confidence:** 5

**Summary:**

This paper presents a theoretical and empirical analysis of how semantic associations emerge in transformer-based language models during early training. By analyzing the leading-term approximation of training gradients, the authors derive closed-form expressions for the model's weight matrices. They show that these weights can be characterized by the composition of three interpretable basis functions—bigram, interchangeability, and context mappings—which collectively explain how transformers acquire semantic structure from data. The theoretical claims are rigorously validated through experiments on both the controlled TinyStories benchmark and large-scale models like Pythia-1.4B, demonstrating strong agreement between the theoretically predicted features and the empirically learned weights.

**Strengths:**

1. The paper starts from the perspective of gradient flow and utilizes the contextual distribution in the text to explain the implied semantic associations within the parameters. This perspective is very novel and provides an in-depth way of understanding the distributional characteristics of the parameters of transformer-based language models.
2. The paper is trained on real text, and the experimental results are highly consistent with the theoretical analysis, demonstrating the reasonableness of its results.

**Weaknesses:**

1. The paper is validated only on the TinyStories dataset, which I consider insufficient for comprehensive verification.
2. The training loss on TinyStories remains very high (greater than 5), indicating that the model's learning on this dataset is inadequate and arguably unsuccessful. The value of discussing parameter characteristics under such inadequate fitting conditions is highly questionable.
3. The description of how theoretical and experimental results are compared in Section 5.2 is difficult to understand. The authors should revise this section and provide more detailed methodological descriptions in the appendix, preferably in an algorithmic format.
4. The paper uses one-hot encoding instead of the more commonly used embedding encoding. The authors should discuss how this distinction affects their theoretical results, particularly whether the presence of an embedding layer under zero initialization would invalidate their theoretical findings.
5. Paper [1] also discusses from a gradient flow perspective how the embedding layer is influenced by semantic distributions under small initialization, and its findings share some similarities with certain results in this paper. I believe the authors should discuss the distinctions between their work and [1].

[1] An Analysis for Reasoning Bias of Language Models with Small Initialization, Forty-second International Conference on Machine Learning.

**Questions:**

See weakness.

---

> ### Author Response · Authors · 2025-11-20
>
> We sincerely appreciate your constructive feedback and insightful comments! We address the questions and comments below in detail.
>
> > `W1`. Validation beyond TinyStories dataset
>
> We would like to clarify that the validation in section 5.2 uses **OpenWebText rather than TinyStories as mentioned in `L427-428`**. Furthermore, we have extended the evaluation using another web data corpus, FineWeb [1]. Results from these extended analyses demonstrate that our theoretical explanation consistently holds across datasets. We have incorporated the **additional results based on FineWeb in `Appendix B`** in our manuscript.
>
> > `W2`. Validity of the weight parameter analysis given the high training loss.
>
> We would like to clarify that the goal of our paper is to explain how the _core, foundational associative features emerge_, which often arises before the model is fully converged to a very low loss. Our theory precisely analyzes when the model constructs the basic token–token association structure (e.g., directional matching, bigram-style dependencies, context-basis features). Our theorem captures this mechanistic process with a clean, realistic analysis.
>
> Accordingly, the empirical setup and **training loss behavior exactly match our theoretical focus**. The training loss remains relatively high because the model is still in the relatively early associative-learning regime. _Even with a higher final loss, the model clearly does learn the foundational structures our theory predicts, and the theoretical leading-term approximation remains strongly aligned with the learned weights_. These results provide a solid scaffold for future theoretical study to analyze more complex setups as well as an alternative perspective to monosemanticity. Furthermore, we provide novel insights on how simple semantic relationships start to form. Being able to successfully describe more complex structures learned at later stages of training is an important problem, and we hope that the current analysis can serve as a framework for future developments.
>
> > `W3`. Writing clarity of Section 5.2
>
> Thank you for the helpful feedback on the clarity of Section 5.2. We have thoroughly expanded this section (`L435-464`) to make it easier to understand and included a step-by-step methodological description.
>
> > `W4`. One-hot vs. Embedding Encoding
>
> We use one-hot encoding because it provides vocabulary space interpretation while maintaining the same expressivity as token embeddings. Any embedding matrix $W_{emb}$ can be absorbed into the other parameters. For example, to compute the logits in the attention layer with $W_{kq}$ as the combined key-query matrix, $(X W_{emb}) W_{kq} (X W_{emb})^\top = X W_{emb} W_{kq} W_{emb}^\top X^\top$. Then, we could compute the same logits without an embedding matrix but with a new key-query matrix equal to $W_{emb} W_{kq} W_{emb}^\top$. As a result, **the presence of an embedding layer with non-zero initialization would not invalidate the findings**.
>
> > `W5`. Distinction from related work
>
> Thank you for the pointer to this highly relevant work! **We have included [2] Yao et al. 2025 in the related work section of our revised PDF.** The key difference in their analysis is that they aim to provide an understanding of **reasoning and memorization through a synthetic task** while our focus is on general semantic associations in natural language data. Furthermore, to analyze model behavior, they consider a two-layer transformer, whereas we consider transformers of varying depth.
>
> [1] Penedo et al. 2024, "The FineWeb Datasets: Decanting the Web for the Finest Text Data at Scale"
> [2] Yao et al. 2025, "An Analysis for Reasoning Bias of Language Models with Small Initialization"

---

> ### Comment · Reviewer_T8ho · 2025-11-26
>
> Thank you very much for the authors' response. Most of my concerns and questions have been adequately addressed, and the current results presented in the paper are already quite solid. Therefore, I maintain a positive stance on this paper.

---

### Official Review · Reviewer_p4Xp · 2025-11-01

**Soundness:** 3
**Presentation:** 3
**Contribution:** 3
**Rating:** 6
**Confidence:** 3

**Summary:**

This paper studies how semantic associations first emerge in attention-based language models by analyzing early training dynamics. Using a leading‑term expansion of the gradients, the authors derive closed‑form approximations for all major weight classes in a self attention only transformer with causal masking and learned relative positional bias.
shows very high cosine similarity between learned weights and their leading‑term predictions, and an analysis of Pythia‑1.4B on OpenWebText shows strong early training alignment that gradually drifts laterr. Qualitative examples illustrate that the basis functions recover intuitive relations like "fish" <--> "pond/lake".

**Strengths:**

* The three basis functions (bigram, interchangeability provide an intuitive, corpus‑linked explanation for what each weight class is learning and how attention and values cooperate early in training


* The writing is very clear

* Near perfect cosine agreement on TinyStories over many steps and reasonable agreement in a non-toy sized model (pythia) support external validity. heatmaps/diagrams are clear

**Weaknesses:**

* Quantitative validation centers on cosine similarity and selected qualitative token lists. Some broader behavioral evaluations or ablations (e.g., other corpora, tokenizations, or stronger baselines) would further strengthen the claims

* Unless I'm missing something, this doesn't hold well for later stages of training because of drift. That's fine, but I figured I'd raise it. I am not entirely sure how much this is a problem for the scope of the paper. Are there analyses on later stages of training that would be interesting but are blocked by this constraint?

* I would like to see some interventional results showing some practicality: E.g., show that the leading term predictors enable training diagnostics or interventions (e.g., early phase monitoring that forecasts later perplexity or feature formation).

* The theoretical result is clean and realistic for early training, and the empirical results are suggestive. I think the thing holding this paper back right now are scope

**Questions:**

N/A

---

> ### Author Response · Authors · 2025-11-20
>
> We sincerely appreciate your constructive feedback and insightful comments! We address the questions and comments below in detail.
>
> > `W1`. Broader evaluations/ablations
>
> Thank you for the helpful suggestion! To address your concern, we have performed the following additional experiments:
>
> 1. **Extended evaluation on Pythia using other corpora**, such as FineWeb [Penedo et al., 2024], another large corpus based on web data (`Appendix B`). We find that the cosine similarity evaluation based on FineWeb is very similar to that based on OpenWebText.
> 2. **Validation under alternative tokenization**, confirming that our theoretical predictions continue to match the learned weights on TinyStories even with a different tokenization scheme, Byte-Pair Encoding (BPE) tokenization (`Appendix B`). We use the BPE tokenization for TinyStories and perform the cosine similarity analysis, and find that after 10 epochs, the cosine similarity for all weights stays above 0.997.
> 3. **Additional analyses of MLP-layer effects and head-level variation**, further illustrating how different components of the model interact with the predicted basis features (`Section 5.2`, page 10).
>
>     a. **MLP** - We perform an analysis comparing the embeddings with and without the MLP output at each layer, and after the first layer, the cosine similarity scores with and without the MLP differ by 0.01-0.1.
>
>     b. **Head-level Variation** - We explored the behavior of specific attention heads, providing a comparison of how quickly heads drift from our leading term expression. We find that earlier layers learn the leading-term features more slowly and that intermediate layers seem to have individual heads specialize faster than other layers. **Tracking whether similarity with the basis functions is high could be a diagnostic for determining if layers/heads are becoming specialized.**
>
> The full results for these new experiments can be found in Section 5.2 and Appendix B in our revised manuscript. **Results from these extended analyses demonstrate that our theoretical explanation consistently holds across diverse setups**.
>
>
> > `W2`. Analysis on later stage dynamics.
> > `W4`. Scope
>
> We agree with your comment that being able to analyze how the weights change at later stages of training would be insightful. However, we believe this limitation is appropriate for the scope of our contribution, for three key reasons:
>
> 1. The goal of the paper is to explain **how the _core, foundational_ associative features emerge**, not to fully characterize all representational refinements.
>     The early-training phase is where the model first builds the foundational token-association structure (e.g., direction matching, bigram-like associations, context basis). Our theorem captures this mechanistic process with a clean, realistic analysis.
>
> 2. Despite drift, the **theoretically predicted features remain strongly expressed well beyond the provable regime**.
> As shown in our experiments, the cosine similarity between learned weights and the leading-term approximation stays extremely high (>0.9 for many epochs and >0.7 even after 100 epochs), indicating that the early-learned features continue to structure the model even after significant training progress. This suggests that our analysis explains not only the strict early stage but also can serve as a structural backbone for longer into training.
>
> 3. Later-stage analyses are interesting but meaningfully distinct--but **attempting to cover both regimes in a single paper would produce a much less focused and less rigorous contribution**.
> We agree that understanding drift is interesting; however, doing so requires addressing multiple fundamental problems, such as higher-order gradient interactions, attention-head specialization, and cross-layer interactions. The challenges, along with the various phenomena seen in models at later stages, suggest that providing a general characterization of model features at these stages will require careful analysis on its own and tools beyond the gradient leading term. We hope that future work will be able to make progress towards characterizing model internals at later stages, and believe our work can be a stepping stone for that.
>
>
> > `W3`. Interventional Results
>
> An intervention or diagnostics based on early stage features would be an exciting direction to explore, and one possible diagnostic (as mentioned in `W1`) would be **tracking whether similarity with the basis functions is high to determine if layers/heads are becoming specialized.** We look into the behavior of individual attention heads across different layers of Pythia in `Figure 7`, and we find that
> 1. Earlier layers learn the leading-term features more slowly.
> 2. Intermediate layers seem to have individual heads specialize faster than other layers.
>
> By comparing attention heads with the leading-terms, an indicator of early general features, we can get a high-level view of how training is progressing.

---

### Official Review · Reviewer_qEwF · 2025-11-06

**Soundness:** 3
**Presentation:** 3
**Contribution:** 3
**Rating:** 8
**Confidence:** 3

**Summary:**

The paper studies how semantic associations between tokens emerge during the early training of language models. Focusing on a self-attention-only architecture, the authors derive the leading-order gradient updates and show that the learned weights can be decomposed using three quantities that reflect the statistics of the training corpus: bigram mapping (correlations between consecutive tokens), intercheangiability mapping (correlations induced by similarity of tokens’ previous-token context distributions), and context mapping (average contextual features of tokens across their occurrences). They validate the theory on small attention-only models - where the predicted leading terms match the learned parameters in the early steps - and then analyze embeddings in Pythia models, finding qualitative alignment with the theory in the initial training phase.

**Strengths:**

- The paper presents an interesting and, to my knowledge, novel blend of mathematical analysis, numerical experiments, linguistic insight, and mechanistic interpretability. The bottom-up approach of deriving representations from first principles of training dynamics is significant.
- The paper analytically derives a leading-order approximation of the weight evolution and provides clear, intuitive interpretations for each component.
- The theory does not rely on synthetic data models. Rather, the basis functions are derived directly from the statistics of the language corpora.
- The paper is well-written and its assumptions are clearly outlined.

**Weaknesses:**

- The primary weakness is that the theory still relies on strong architectural assumptions. It is derived only for self-attention-only transformers. This analysis does not account for the impact of MLP layers, which are important components of present models. While this is a limitation, it is an understandable simplification necessary to make the analysis tractable, and future work might build upon this to relax these assumptions.

**Questions:**

- Th. 4.1 provides formal bounds on the error between the learned weights and their leading-term approximations. How tight are these error bounds in practice? For instance, in the TinyStories experiment.

---

> ### Author Response · Authors · 2025-11-20
>
> We sincerely appreciate your positive feedback and insightful comments! We address the questions and comments below in detail.
>
> > `W1`. The impact of MLP Layers
>
> We fully agree that understanding the impact of MLP layers is important. In response to this comment, we conducted additional analysis on how MLP layers interact with the basis functions by comparing the structure of embeddings with and without the MLP output. The results are shown in the revised `Section 5.2`. We found that the embeddings without the MLP output have a similar structure to those with the MLP output, and after the first layer, the cosine similarity scores with and without the MLP differ by 0.01-0.1.
>
> As you have noted, the high complexity of MLP transformations makes deriving clean theoretical statements challenging, especially across different checkpoints. Nevertheless, our empirical results on Pythia indicate that **embeddings taken from the residual stream _after_ both the value mapping and the MLP layer still preserve the token-level structure predicted by the value-mapping analysis.**
>
> > `Q1`. The error bound tightness
>
> The tightness of the bound grows weaker over training, and as seen by the duration for which the weights resemble the leading terms, the error bound could be reduced by a significant factor. The main challenge in doing so comes from approximating the dynamics, particularly due to limitations in current theoretical tools for the softmax function. Providing a closed-form approximation of the softmax of a vector, even in the forward pass, can be challenging and would require developing new approximation techniques. This challenge is one of the main reasons existing work often considers **Mean Squared Error** [1], **simplified architectures** [2], or **highly structured data** [3]. However, while our bound may not be optimal, we believe that the leading terms themselves provide new insights into the structure of model features for a significant portion of training. We hope future work will make progress toward more precise dynamics and characterizing model internals at later stages, which we believe our work can serve as a stepping stone for.
>
> [1] Kim, Juno, and Taiji Suzuki. "Transformers learn nonlinear features in context: Nonconvex mean-field dynamics on the attention landscape." arXiv preprint arXiv:2402.01258 (2024).
>
> [2] Edelman, Ezra, et al. "The evolution of statistical induction heads: In-context learning markov chains." Advances in neural information processing systems 37 (2024).
>
> [3] Nichani, Eshaan, Alex Damian, and Jason D. Lee. "How transformers learn causal structure with gradient descent." arXiv preprint arXiv:2402.14735 (2024).

---

### Official Review · Reviewer_bKRV · 2025-11-11

**Soundness:** 3
**Presentation:** 4
**Contribution:** 4
**Rating:** 8
**Confidence:** 4

**Summary:**

This paper addresses a critical gap in transformer interpretability: how semantic associations (e.g., "bird"-"flew", "country"-"capital") emerge during training of attention-based language models (LLMs) on natural language data. Unlike prior work that relies on synthetic data, simplified architectures, or non-standard training, the authors ground their analysis in realistic settings (natural text distributions, standard transformers with positional encoding, and standard next-token prediction loss). Their key technical innovation is leveraging a gradient leading-term approximation to derive closed-form expressions for transformer weights in the early training stage.

The authors show that all transformer weights (output, value, query-key, positional encoding) can be characterized as compositions of three interpretable basis functions: (1) bigram mapping (captures next-token dependencies), (2) token-interchangeability mapping (reflects functional similarity, e.g., synonyms), and (3) context mapping (encodes long-range prefix-suffix co-occurrence). They validate this theory empirically: on a 3-layer transformer trained on TinyStories, learned weights maintain cosine similarity ≥0.9 with theoretical predictions even beyond early training; on real-world LLMs (Pythia-1.4B trained on OpenWebText), early-stage activations and attention weights strongly align with the proposed basis functions.

**Strengths:**

1. Realism of the theoretical setup: By retaining critical components of practical transformers (positional encoding, causal masking, residual streams) and using natural text, the paper avoids the over-simplifications that limit the generalizability of prior work.

2. Mechanistic interpretability: The basis functions provide a causal explanation for semantic association (e.g., the value matrix combines context and bigram mappings to encode long-range semantics), rather than just correlational observations.

3. Strong empirical validation: The experiments are comprehensive, testing the theory on both small controlled models (TinyStories) and large real-world LLMs (Pythia-1.4B), with quantitative (cosine similarity) and qualitative (token correlation examples, Figure 5) evidence.

4. Foundational value for future work: The closed-form expressions and basis functions can serve as a starting point for further research (e.g., diagnosing bias in LLMs by analyzing deviations from the theoretical bigram/context mappings, or designing more interpretable architectures).

**Weaknesses:**

1. Limited analysis of later training stages: While the paper notes that theoretical features persist beyond early training (e.g., cosine similarity ≥0.7 after 100 epochs in TinyStories), it does not explore why or how weights drift from the leading terms. Understanding this drift (e.g., whether it corresponds to higher-level semantic learning) would strengthen the theory’s completeness.

2. Multi-head attention and MLP layers are understudied: The Pythia experiment adapts the analysis to multi-head attention by averaging attention heads, but it does not explore how individual heads or MLP layers interact with the proposed basis functions. For example, do some heads specialize in bigram mappings while others focus on context?

3. Lack of causal interventions: The empirical validation relies on correlational measures (cosine similarity, covariance), not causal tests (e.g., ablating the bigram mapping component of weights to see if next-token prediction degrades). Causal interventions would more strongly confirm that the basis functions are necessary for semantic associations.

**Questions:**

1. Drift in later training stages: You note that weights drift from leading terms as training progresses (Figure 6). Can you characterize this drift? For example, does it correspond to learning higher-level semantics (e.g., syntax, pragmatics) that build on the initial basis functions, or does it reflect overfitting to idiosyncrasies in the data?

2. Multi-head attention specialization: In Pythia, you average attention heads to compute token correlations, but prior work shows heads specialize in different tasks. Do individual heads align with specific basis functions (e.g., some heads focus on bigram mappings, others on context)? If so, how does this specialization emerge?

3. Causal validation: Your analysis uses correlational measures (cosine similarity) to link learned weights to theoretical terms. Have you tried causal interventions (e.g., modifying the bigram mapping component of the output matrix and measuring changes in next-token prediction accuracy for bigram-dependent pairs like "bird"-"flew")? Such tests would strengthen the claim that the basis functions are functional, not just correlational.

---

> ### Author Response · Authors · 2025-11-20
>
> We sincerely appreciate your positive feedback and insightful comments! We address the questions and comments below in detail.
>
> > `W1`/`Q1`. Analysis on later training stage and characterization of weights drift.
>
> We fully agree with your comment that being able to analyze how the weights change at later stages of training would be insightful. However, **such an analysis is primarily bottlenecked by the limited tools to analyze the softmax function.** In particular, providing a closed-form approximation of the softmax of a vector even in the forward pass can be challenging. Therefore, analyzing long-range training dynamics would require developing new techniques beyond the gradient leading term approximation. This challenge is one of the main reasons existing work often considers Mean Squared Error [1], simplified architectures [2], or highly structured data [3].
>
> Regarding the characterization of the drift across training, **it likely corresponds to learning higher-level semantics rather than overfitting**, since we know that the model becomes more capable at later stages and forms structures such as induction heads and representations of higher-level concepts [4, 5]. The various observations about model behavior at later stages suggest that providing a general characterization of model features at these stages will require careful analysis.
>
> We hope that future work will be able to make progress towards characterizing model internals at later stages and believe these to be exciting directions.
>
> [1] Kim, Juno, and Taiji Suzuki. "Transformers learn nonlinear features in context: Nonconvex mean-field dynamics on the attention landscape." (2024).
>
> [2] Edelman, Ezra, et al. "The evolution of statistical induction heads: In-context learning markov chains." (2024).
>
> [3] Nichani, Eshaan, Alex Damian, and Jason D. Lee. "How transformers learn causal structure with gradient descent." (2024).
>
> [4] Belrose, Nora, et al. "Neural networks learn statistics of increasing complexity." (2024).
>
> [5] Wang, George, et al. "Loss landscape geometry reveals stagewise development of transformers." (2024).
>
> > `W2`/`Q2`. Multi-head attention specialization and MLP layer analysis.
>
> We appreciate the insightful questions and have performed further analysis.
>
> **(1) MLP.** We fully agree that understanding the impact of MLP layers is important and have conducted additional analysis on how MLP layers interact with the basis functions. Our empirical results on Pythia based on embeddings from the residual stream, which combine both the value/output mapping and the MLP layer, **suggest that the MLP maintains the structure between tokens that we would expect from only having the value/output mapping.** To further verify this, we perform the analysis using embeddings without the MLP output, and the result is shown in the revised `Section 5.2`. The results are similar to those with the MLP output, and after the first layer, the cosine similarity **scores with and without the MLP differ by 0.01-0.1.**
>
>
> **(2) Multi-head attention**. Based on the theory where each layer receives similar gradient signals, we expect attention heads to be less specialized at early training stages and for all of them to roughly correspond to the context and bigram composition. We likely would not see specialization in the form of certain heads specializing in bigram mappings while others specialize in context mapping, since we expect a composition of the two mappings to be learned, and having these in parallel would make this more challenging. Instead, since heads have different initializations in practice, they may specialize at different rates or in different directions after learning the initial semantic features. To validate this, we explored the behavior of specific attention heads in `Figure 7` in our revised manuscript, providing a comparison of how quickly heads drift from our leading term expression over time. **We find that earlier layers learn the leading-term features more slowly and that intermediate layers seem to have individual heads specialize faster than other layers.**
>
> > `W3`/`Q3`. Causal validations
>
> We perform a simple causal validation on the TinyStories setting to see how much the leading term features contribute to next-token prediction **by comparing the loss on the dataset before and after removing the leading term component**. We provide details and results in `Appendix B` and summarize the results below.
> 1. We find that removing the leading term from the output matrix affects the model's ability the most, while removing the leading term from the attention weights affects it the least. This **aligns with our theory, which shows that the output layer gets the largest order updates** while the attention weights get the smallest order updates.
> 2. We see that for the output and value weights, removing the leading term can **increase the loss from 5.35 to 8.29** and 6.53, respectively, suggesting that the basis functions are functional.

---

### Author Response · Authors · 2025-11-20

We would like to thank the reviewers and chairs for their time and effort in providing thoughtful feedback on our work.

This paper addresses a critical question underlying LLM research today: _how do semantic associations emerge during the training of attention-based language models_? We contribute a foundational framework grounded in a leading-term approximation of training gradients, revealing the composition of three interpretable basis functions—bigram, interchangeability, and context mappings—which collectively explain how transformers start to acquire semantic structure from data.

Across five reviewers, there was unanimous recognition of the work’s novelty, rigorous theoretical analysis, and strong alignment between theory and empirical results:

1. **Offers strong foundational value**:
- "_The bottom-up approach of deriving representations from first principles of training dynamics is significant._" --- Reviewer qEwF
- "_The closed-form expressions and basis functions can serve as a starting point for further research._" --Reviewer bKRV
2. **Introduces a novel and interesting technical approach** with **realistic setup**:
- "_The paper presents an interesting and, to my knowledge, novel blend of mathematical analysis, numerical experiments, linguistic insight, and mechanistic interpretability._" --- Reviewer qEwF
- "..._This perspective is very novel and provides an in-depth way of understanding the distributional characteristics of the parameters of [..] language models._" --- Reviewer T8ho
- "_Unlike much prior theoretical work that relies on synthetic data or heavily simplified models, this analysis is grounded in a more realistic setup._" --- Reviewer XVTW
3. **Strong alignment between theory and empirical analysis**:
- "_The paper is trained on real text, and the experimental results are highly consistent with the theoretical analysis, demonstrating the reasonableness of its results._" ---Reviewer T8ho
4. **Clearly written** with assumptions clearly outlined. --- Reviewers qEwF and p4Xp

### Responses and changes in the manuscript

- **Broader empirical validation** (p4Xp, XVTW): We extend our Pythia evaluation to FineWeb, validate results on TinyStories using a BPE tokenization (10K vocab), and update all Pythia experiments to use the full vocabulary (50,304 tokens).
- **MLP and attention head analysis** (bKRV, qEwF, XVTW): We extend our analysis to compare the structure of embeddings with and without MLP layers. We have performed additional analysis on individual attention heads to understand how specialization emerges over time and across layers.
- **Writing clarity and completeness** (T8ho): We revised Section 5.2 for clarity and described the method in a step-by-step manner, and added further discussion on previous work in the related work section.

---

### Meta-Review · Area_Chair_yVzX · 2026-01-05

**Summary:**

This paper presents a clear and well-motivated theoretical and empirical study that contributes to a better understanding of how semantic associations emerge during the training of attention-based language models. The methodology is technically sound, the theoretical analysis is carefully developed, and the empirical results are consistent with the paper’s claims. The reviewers agree that the work is solid and provides meaningful insights, with no fundamental flaws identified. Overall, I find no serious concerns that would affect acceptance.

**Reviewer Concerns:**

The main concerns raised by the reviewers were: (i) the applicability of the theoretical results primarily to the early phase of training, (ii) the focus on a limited set of network architectures, and (iii) the scope of the experimental evaluation. These points were well addressed in the rebuttal. The authors clarified the intended scope of the theory, provided additional discussion to contextualize its relevance beyond early training, and supplied supplementary experimental results that strengthen the empirical support. As a result, no major concerns remain outstanding.

**Reviewer Scores:**

Concerns regarding limited experimental coverage and the applicability of the theory were satisfactorily addressed in the rebuttal. The additional results and clarifications resolve the reviewers’ main reservations. All reviewers would likely retain their current scores, and given the moderate confidence level (confidence score 3) of Reviewer p4Xp, it is plausible that this reviewer would increase their score following the discussion.

---

### Decision · Program_Chairs · 2026-01-26

Accept (Oral)